# A bitopic agonist bound to the dopamine 3 receptor reveals a selectivity site

Sandra Arroyo-Urea [1,2], Antonina L. Nazarova [3,4], Ángela Carrión-Antolí[1,2], Alessandro Bonifazi [5], Francisco O. Battiti[5], Jordy Homing Lam [3,4], Amy Hauck Newman [5], Vsevolod Katritch [3,4,6] & Javier García-Nafría [1,2] ✉

Although aminergic GPCRs are the target for ~25% of approved drugs, developing subtype selective drugs is a major challenge due to the high sequence conservation at their orthosteric binding site. Bitopic ligands are covalently joined orthosteric and allosteric pharmacophores with the potential to boost receptor selectivity and improve current medications by reducing off-target side effects. However, the lack of structural information on their binding mode impedes rational design. Here we determine the cryo-EM structure of the $hD_3R:G\alpha_O\beta\gamma$ complex bound to the $D_3R$ selective bitopic agonist FOB02-04A. Structural, functional and computational analyses provide insights into its binding mode and point to a new TM2-ECL1-TM1 region, which requires the N-terminal ordering of TM1, as a major determinant of subtype selectivity in aminergic GPCRs. This region is underexploited in drug development, expands the established secondary binding pocket in aminergic GPCRs and could potentially be used to design novel and subtype selective drugs.

While G protein-coupled receptors (GPCRs) form the largest family of drug targets, accounting for more than a third of FDA-approved drugs[1], developing subtype-selective drugs is a major challenge. This is especially true for aminergic GPCRs, which include 42 receptors (dopamine, serotonin, adrenaline, histamine, acetylcholine, and trace amine receptors) with high sequence similarity. In the most closely related aminergic receptor subtypes, sequence identity often exceeds 80% of the orthosteric binding site (OBS) residues. Such conservation supports neurotransmitter promiscuity[2] between subtypes, but results in undesired off-target side effects of drugs that only bind in the OBS[3]. Controlling drug selectivity for aminergic receptors has the potential to improve current therapies and it could be achieved by the design of bitopic molecules[4–6]. These are ligands generated by covalently joining two pharmacophores, a primary pharmacophore (PP), that usually targets the OBS, and a secondary pharmacophore (SP), that targets an allosteric or secondary binding pocket (SBP) generally divergent in sequence and/or structure within the target receptor[4–7]. Hence, bitopic molecules have been proposed to have a separate "message-address" system wherein an agonist/antagonist, the message, is linked to a pharmacophore binding to the SBP, which contains the address[5,8]. Indeed, several bitopic compounds with enhanced receptor selectivity have been developed for GPCRs[9–12]. Overall, bitopic ligands present a rational approach to develop molecules with enhanced functionality and selectivity, however there is scarce structural information on their binding modes and development relies heavily on structure-activity relationships together with computational and synthetic strategies[13–16]. Here we aim to understand the molecular basis of a selective bitopic molecule that distinguishes between two closely related aminergic GPCRs, the human dopamine $D_2$ receptor ($D_2R$) and dopamine $D_3$ receptor ($D_3R$). These receptors share 78% sequence identity at the transmembrane segment and 100% identity at the OBS, making their pharmacological distinction a notoriously hard challenge[17,18]. $D_2R$ and

[1]Institute for Biocomputation and Physics of Complex Systems (BIFI), University of Zaragoza, Zaragoza, Spain. [2]Laboratory of Advanced Microscopy (LMA), University of Zaragoza, Zaragoza, Spain. [3]Department of Quantitative and Computational Biology, University of Southern California, Los Angeles, CA, USA. [4]Center for New Technologies in Drug Discovery and Development, Bridge Institute, Michelson Center for Convergent Biosciences, University of Southern California, Los Angeles, CA, USA. [5]Medicinal Chemistry Section, Molecular Targets and Medications Discovery Branch, National Institute on Drug Abuse – Intramural Research Program, National Institutes of Health, 333 Cassell Drive, Baltimore, Maryland, USA. [6]Department of Chemistry, University of Southern California, Los Angeles, CA, USA. ✉e-mail: jgarcianafria@unizar.es

$D_3R$ differ in brain distribution and signaling properties and are both targeted by current antipsychotics and drugs for the treatment of neurological diseases (such as Parkinson's disease[19,20]). Although agonists with some selectivity exist[20,21], new subtype selective molecules are likely to help understand their physiological role as well as provide leads for improved therapeutics. Indeed, selective activation of $D_3R$ may yield neuroprotective effects in the treatment of Parkinson's disease, hence harboring potential in the treatment of neurodegeneration[21,22].

In this work, we determine the cryo-electron microscopy (cryo-EM) structure of the human $D_3R$ bound to a bitopic and full agonist (FOB02-04A) and coupled to a $G\alpha_O\beta\gamma$ heterotrimer. Together with functional assays, mutagenesis, docking studies, and molecular dynamics (MD) simulations, we determine the binding mode and basis for the $D_3R$ selectivity of this compound. The bitopic molecule occupies the OBS and protrudes towards the outside of the ligand binding pocket to contact an allosteric site at the extracellular vestibule of $D_3R$ formed by TM2-ECL1 and TM1. This region is of high sequence and structural variability and expands the established aminergic SBP, opening new avenues to develop subtype-selective bitopic drugs, potentially across other aminergic GPCRs.

## Results

### Overall cryo-EM structure of the hD₃R:Gα_Oβγ:scFv16 bound to a bitopic ligand

The $D_3R:G_O$ heterotrimer:FOB02-04A complex was produced by co-expressing the $hD_3R$ ($L^{3.41}W$ mutation following Ballesteros–Weinstein numbering[23]), the dominant negative $G\alpha_O$ subunit[24], $G\beta_1$ and $G\gamma_2$ in insect cells (see "Methods") (Supplementary Fig. 1A). The $D_3R^{L3.41W}$ was previously used in structural studies[25] and was validated in this work using cellular BRET assays[26], where it displayed a virtually identical ligand-induced activation as the wild-type $D_3R$ (Supplementary Fig. 1B). The bitopic FOB02-04A was synthesized as previously described[9] and was added before complex solubilization from insect cell membranes. The scFv16[27] (which binds to the $G\alpha_O$:Gβ interface) was incorporated prior to size exclusion chromatography. The structure of the complex was then solved by single-particle cryo-EM (Fig. 1 and Supplementary Fig. 2). Positioning the ligand binding pocket at the center of the cryo-EM box improved the resolution at the $D_3R$ extracellular region (Methods and Supplementary Fig. 3), and allowed to classify two cryo-EM models containing two FOB02-04A conformations – Conformation A (to a global resolution of 3.05 Å) and B (global resolution of 3.09 Å), which mainly differed in the position of the bitopic SP and residues around the SBP (Fig. 1 and Supplementary Fig. 2). We will initially focus on Conformation A unless otherwise stated since Conformation B was concluded to be a non-productive antagonistic conformation (see below). Both final cryo-EM maps were of sufficient quality to build confidently the $D_3R$, the Gαβγ proteins, the scFv16, and the bitopic FOB02-04A ligand (Supplementary Fig. 4 and Supplementary Table 1). Both $D_3R$ conformations were built from residues $H29^{1.32}$ (Conformation A)/$Y32^{1.35}$(Conformation B) to C400 with missing residues for intracellular loop 3 (ICL3) (missing residues including $I223^{5.73}$ to $R323^{6.29}$). No cholesterol (or cholesterol hemisuccinate) or lipid molecules were found around the transmembrane part of the receptor, consistent with previously reported structures of the $D_2R$[28–30] and $D_3R$[25,30,31] and in contrast to $D_1R$, $D_4R$, and $D_5R$ where cholesterol was bound to the transmembrane segment[30,32,33].

### Activation mechanism and G_O coupling of the D₃R bound to FOB02-04A

The $D_3R:G\alpha_O\beta\gamma$:FOB02-04A displays the characteristic structure of a GPCR:G protein complex, with resemblance to the previously determined structures of $D_3R$ coupled to a $G_i$ heterotrimer[30,31] (e.g., RMSD of 1.036 Å for 1022 Cα for the pramipexole bound structure (PDB 7CMU). No major conformational changes were found at the $D_3R$ when comparing its structure when bound to pramipexole (PDB 7CMU), PD128907 (PDB 7CMV), rotigotine (PDB 8IRT) or FOB02-04A (0.535 Å RMSD over 253 Cα in the pramipexole bound as example) aside from the ordering of the extracellular region of TM1 (see below). The $D_3R$ activation induced by FOB02-04A follows the canonical conformational changes[34], i.e., a downward shift of the toggle switch $W342^{6.48}$, a conformational change of the PIF ($I118^{3.40}$, $F338^{6.44}$), DRY ($D127^{3.49}$, $R128^{3.50}$, $Y129^{3.51}$) and NPxxY ($N379^{7.49}$, $P380^{7.50}$, P $Y383^{7.53}$) motifs, which end up with an ~ 9 Å outward swing of the cytoplasmic end of TM6 and inward movement of TM7 toward the core of the receptor as compared to the inactive state[25] (Supplementary Fig. 5). The coupling of the $G_O$ heterotrimer to the $D_3R$ occurs through two interfaces: a first major interface located between the $G\alpha_O$ C-terminal α5, that engages mainly with the intracellular part of TM3, TM5 and TM6 of the $D_3R$ ($I344^{G.H5.16}$, $L348^{G.H5.20}$, $C351^{G.H5.23}$, $L353^{G.H5.25}$, and $Y354^{G.H5.26}$ in $G\alpha_O$ packing against $R128^{3.50}$, $A131^{3.53}$, $V132^{3.54}$, $I211^{5.61}$, $L215^{5.65}$, $R218^{5.68}$, $R222^{5.72}$, $R323^{6.29}$, $K326^{6.32}$, $A327^{6.33}$ and $M330^{6.36}$ in $D_3R$) with contributions from TM7 and TM2 (Fig. 2A). Of note, from MD simulations spanning five independent 0.6 μs runs of the $D_3R$ bound to FOB02-04A and coupled to $G\alpha_O\beta\gamma$ within a membrane bilayer, alternating salt bridge interactions occurred between $D341^{G.H5.13}$ (superscript denotes CGN numbering system[35]) of the $G\alpha_O$ C-terminal α5 and the guanidinium groups of $R218^{5.68}$ and $R222^{5.72}$ in $D_3R$ (Supplementary Fig. 6). A second interface is located at the intracellular loop 2 (ICL2), which makes interactions in a pocket formed by the $G\alpha_O$ N-terminal helix, the C-terminal α5 and the loops connecting the β-strands. The interaction is also held together by unspecific electrostatic charges between the receptor and the Gα protein conserved among $G_{i/O}$ coupled receptors[30,31].

The $D_3R$ has been shown to couple preferentially to $G_O$ compared to $G_i$[36]. We have validated the $D_3R$ $G_O/G_i$ preference by using cellular BRET assays in HEK293T cells[26], which displays a ~ 135-fold difference in potency between $G_O$ and $G_i$ (Fig. 2E). The current $D_3R:G_O$ structure allows us to compare it with the previously determined $D_3R:G_i$ complex to search for potential differences that could explain such $D_3R$ coupling preference (Fig. 2C, D). Overall, both structures exhibit a similar interface area, with $D_3R$-$G_O$ having only a slightly lower buried surface area than $D_3R:G_i$ (959.4 Å² and 1051.8 Å² for $G_O$ and $G_i$ coupled $D_3R$, respectively). However, a smaller interface area is usually seen in $G_O$ vs $G_i$ couplings irrespective of selectivity[37,38]. In addition, both structures present a similar outward swing in TM6 irrespective of $G_O$ or $G_i$ coupling (Fig. 2C), in line with previous observations of the same receptor coupled to different Gα proteins keeping the magnitude of TM6 outward swing[39,40]. However, differences occur when looking at the C-terminal α5 interactions of $G_O$ vs $G_i$. In the case of $G\alpha_O$, the terminal $Y354^{G.H5.26}$ points toward TM5, in contrast to its equivalent $F354^{G.H5.26}$ in $G\alpha_i$, which is sandwiched between $R323^{6.29}$ and $K345^{G.H5.17}$ (this residue is specific for $G\alpha_i$, $A345^{G.H5.17}$ in $G\alpha_O$) (Fig. 2C). Furthermore, previous studies suggested that native ICL contacts are essential to achieve $G_O$ selectivity in $D_3R$[36,41]. Structural differences were also found at the interaction made by ICL2 where, in $G_O$, $Q139^{34.54}$ moves away from the α5 of $G_O$ to interact with $K32^{G.hnsl.03}$ in the αN (Fig. 2D). Such interaction was further confirmed in MD simulations, whose interacting distance remained constant along the five trajectories spanning 0.6 μs each (Supplementary Fig. 6). In addition, the interaction between $Q144^{ICL2}$ and $E28^{G.HN.52}$ in the $G_i$ αN is lost when coupled to $G_O$ due to a replacement of $E28^{G.HN.52}$ by isoleucine as well as the slight difference in the positioning of $G_O$ with respect to the receptor. In order to understand the residues responsible for the $G_O$ selectivity at the $D_3R$ we identified all residues involved at the $D_3R:G\alpha$ protein interface which differed between $G_O$ and $G_i$ and we reverted them one at a time to $G_i$ over the $G\alpha_O$ background, which involved $I28E^{G.HN.52}$, $N194L^{G.s2s3.02}$, $V334F^{G.H5.06}$, $G350D^{G.H5.22}$, and $Y354F^{G.H5.26}$ (Fig. 2F and Supplementary Fig. 5). Overall, only mutation of residue $G350^{G.H5.22}$, located at the C-terminal α5, had a significant impact on its own and

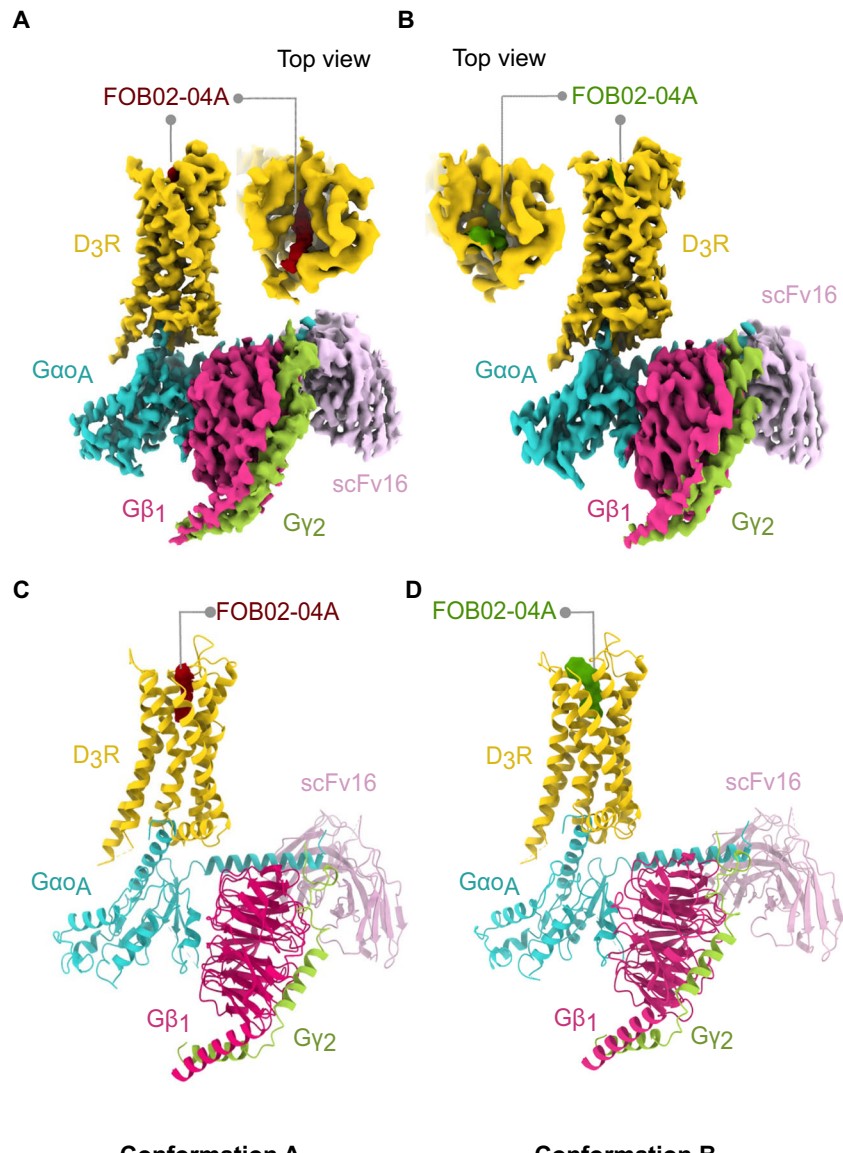

**Conformation A**

**Conformation B**

**Fig. 1 | Overall cryo-EM reconstruction of the $D_3R$-$G_O$:FOB02-04A complex.**
**A, B** Cryo-EM maps for the $D_3R$-$G_{OA}$:FOB02-04A complex in Conformation A (**A**)
and B (**B**) are shown with an inset into the ligand binding site from the top view.
Cryo-EM density is colored according to subunit with the bitopic ligand colored in
red (Conformation A) and green (Conformation B). **C, D** Coordinates for Con-
formation A (**C**) and B (**D**) for both complexes are shown as cartoons and colored by
subunit with the bitopic ligand colored in red (Conformation A) and green
(Conformation B).

hence this residue is the most determinant of the $D_3R$:$G_O$ selectivity
(Fig. 2F). $G350^{G.H5.22}$ from $G\alpha_O$ packs against ICL2, which in $G\alpha_i$ the
bulkier $G350D^{G.H5.22}$ substitution might present steric strains, hindering
$G_i$ coupling. In fact, MD simulations suggest that a second rotamer of
$H140^{34.55}$ is sampled that occupies the gap left at $G350^{G.H5.22}$, and such
rotamer would clash with the $G350D^{G.H5.22}$ substitution in $G\alpha_i$, all
together promoting the coupling to $G_O$ vs $G_i$ (Fig. 2B and Supple-
mentary Fig. 5). Furthermore, in $D_2R$ (a receptor without $G\alpha_{i/O}$ selec-
tivity) ICL2 is displaced outward from the receptor core, yielding a
wider cavity and posing no steric restrictions to either $G\alpha_O$ or $G\alpha_I$
coupling at this position (Fig. 2B). Overall, ICL2 seems to play a relevant
role in determining the $G\alpha_{i/O}$ selectivity at the $D_3R$.

### The binding mode of the bitopic agonist FOB02-04A at $D_3R$

Bitopic molecules are composed of a PP (binding at the OBS), an SP
(binding at the allosteric site), and a linker. FOB02-04A is a full agonist
bitopic molecule composed of a non-catechol PP (based on PF592,379,
an aminopyridinyl-based scaffold), an SP with an indole-amide group,

and a (1$R$,2$S$)-cyclopropyl linker moiety whose chirality has been
optimized for ligand binding and selectivity[9,42,43] (Fig. 3A). The cryo-EM
density allowed modeling of the three components of the bitopic
ligand. Unlike other agonists, which target the bottom of the pocket
exclusively, FOB02-04A binds to the OBS and runs along a narrow
channel towards the allosteric site in the extracellular vestibule,
interacting with residues from TM1-3 and TM5-7 (Fig. 3B–D). The SP of
FOB02-04A is found protruding out of the tight channel to bind in the
extracellular vestibule of $D_3R$, occupying most of the ligand binding
pocket, in contrast to pramipexole which only occupies 23% of the
pocket volume (Fig. 3B, C). Each component of the bitopic molecule
(PP, linker, and SP) occupies a different region within the $D_3R$ pocket,
overall defined by a combination of hydrophobic and polar interac-
tions, as described in Fig. 3E.

The PP pocket at the OBS is defined by strong salt bridge inter-
actions with $D110^{3.32}$, and a cavity formed by $S196^{5.46}$, $F345^{6.51}$, $F346^{6.52}$,
$W342^{6.48}$, $V111^{3.33}$, $T115^{3.37}$ and $I183^{45.52}$, with an additional weak H-bond
with $S192^{5.42}$ (Fig. 3). To correlate structural information with

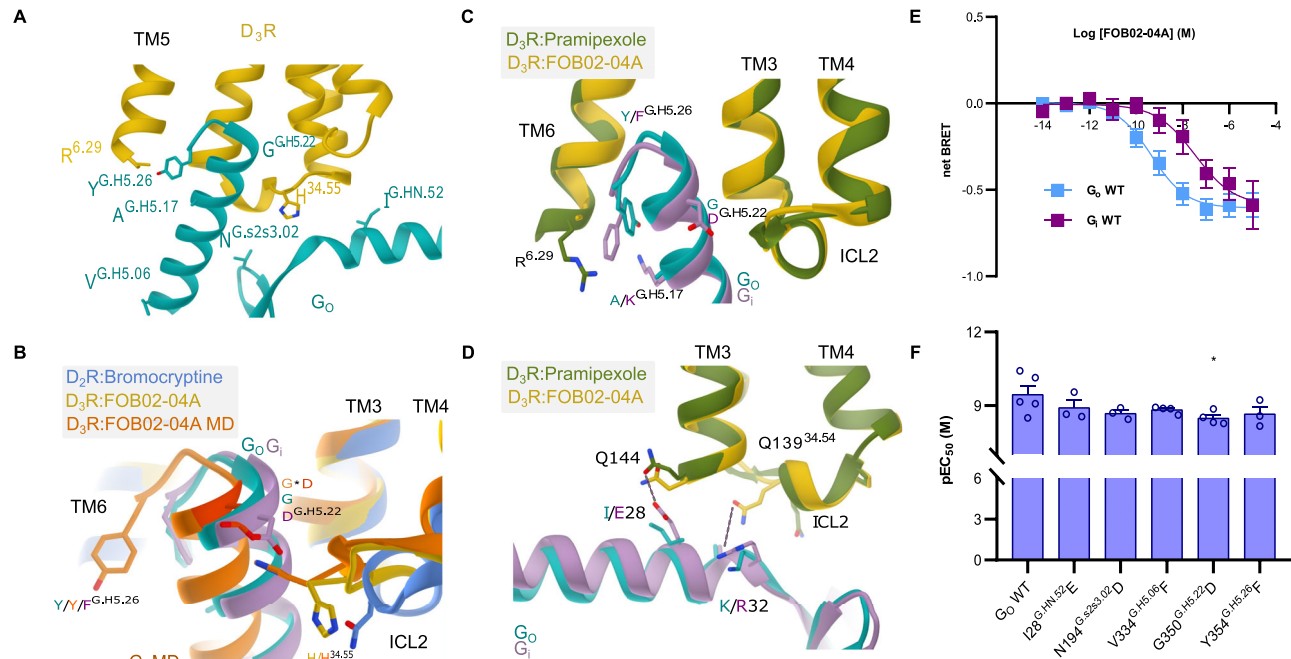

**Fig. 2 | Coupling of D$_3$R to G$_O$ heterotrimer. A** Interaction details of the D$_3$R:G$_O$ interface when bound to FOB02-04A. Cryo-EM density of the C-terminal α5 is shown as mesh. **B** Interaction details of α5 interaction of G$_O$ (turquoise, cryo-EM; orange, MD simulations) and G$_i$ (violet, PDB 7JVR) with ICL2 of D$_3$R (yellow, cryo-EM structure; orange, MD simulation) and D$_2$R (blue, PDB 7JVR). Mutation at residue G350$^{G.H5.22}$ to D in G$_O$-MD is highlighted in dark orange and shown with an asterisk. **C** Comparison of the C-terminal α5 interaction of G$_i$ (PDB 7CMU) and G$_O$ (D$_3$R-G$_O$:FOB02-04A). **D** Interaction details of N-terminal G$_O$ (turquoise) and G$_i$ (violet) with ICL2 of D$_3$R. **E** Concentration-response curve of D$_3$R upon G$_{OA}$ and G$_{i1}$ activation by FOB02-04A using the TRUPATH assay. pEC$_{50}$ values are means ± SEM of five independent experiments performed in technical triplicate. **F** pEC$_{50}$ values for D$_3$R in response to G$_{OA}$ mutants activation by FOB02-04A using the TRUPATH assay. Data are presented as means ± SEM of three (G$_{OA}$ I28$^{G.HN.52}$E, N194$^{G.s2s3.02}$D, Y354$^{G.H5.26F}$), four (G$_{OA}$-V334$^{G.H5.06}$F, G350$^{G.H5.22}$D,) and five (G$_{OA}$ WT) independent experiments performed in technical triplicate. *$p < 0.05$ (one-way ANOVA with Dunnett post hoc analysis) for G350$^{G.H5.22}$D ($p = 0.048$). All source data within this figure is provided as a Source Data file.

functional activity, most of the residues involved in ligand binding were mutated to alanine, following quantification of their surface expression and measurement of their ligand-induced activation using functional BRET assays in HEK293T cells[26] (see Methods and Supplementary Fig. 7). At the OBS there were critical residues which showed no detectable activity when mutated to alanine such as the conserved D110$^{3.32}$, which forms a stable charge interaction with almost all agonists in aminergic receptors, and W342$^{6.48}$, the conserved toggle switch residue at the bottom of the OBS pocket that is essential for signaling. In addition, I183$^{45.52}$, which sandwiches the ligand from the extracellular side (ECL2), V111$^{3.33}$, and T115$^{3.37}$ had a significant impact on agonist potency when mutated (Fig. 3H, I). V111$^{3.33}$ is specifically relevant for FOB02-04A since its mutation does not have an impact on the D$_3$R-induced activation by pramipexole, rotigotine, and PD128907[30,31]. In turn, T115$^{3.37}$ is relevant for FOB02-04A and pramipexole in contrast to PD128907 and rotigotine. Finally, an agonist interaction with S192$^{5.42}$ is found within most aminergic receptor-agonist pairs, however, it seems to be less important for FOB02-04A binding (Fig. 3 and Supplementary Fig. 7). This is in line with non-catechol agonists not relying heavily on S192$^{5.42}$ for binding and activation[44] (also observed for pramipexole[31]). A conserved hydrophobic pocket between T369$^{7.39}$ and F345$^{6.51}$ is efficiently occupied by the rotigotine, pramipexole and PD128907 propylamine group, while it is barely occupied by a methyl group by FOB02-04A (Supplementary Fig. 8). This may explain the lack of effect of F345$^{6.51}$A upon activation by FOB02-04A and suggests that a larger hydrophobic group at this position might improve its binding.

The linker component of the FOB02-04A, which connects PP and SP, interacts with residues at the established SBP in aminergic receptors[3,13], an unexploited region in pramipexole and PD128907 but occupied by the propylthiophene group in rotigotine[30]. The pocket is formed by residues V86$^{2.61}$, F106$^{3.28}$, T369$^{7.39}$, and Y373$^{7.43}$ and has been proposed to have different plasticity among dopamine receptors, and hence a source for ligand specificity[30]. In the case of FOB02-04A, three residues showed a significant reduction in activity when mutated to alanine: Y373$^{7.43}$, F106$^{3.28}$, and V86$^{2.61}$. Y373$^{7.43}$A showed non-detectable activity, and, although this residue is known to be relevant for maintaining the D110$^{3.32}$ geometry to make the conserved charged interactions with agonists, its mutation does not have such a pronounced effect on the activity of pramipexole, dopamine and PD128907[31] as it has on the activity of rotigotine or FOB02-04A. This suggests a role for this residue in the binding and/or function of the bitopic molecule to the receptor, in addition to its known role with D110$^{3.32}$. In addition, the alanine mutation of F106$^{3.28}$ and V86$^{2.61}$ showed reduced efficacy. This is likely to be FOB02-04A specific since V86$^{2.61}$A did not reduce efficacy upon pramipexole activation[31]. Overall, the linker connecting the PP and SP has an active role in the D$_3$R selective binding and function of FOB02-04A and its related bitopic analogs[31].

Finally, the FOB02-04A SP binds in a groove-shaped pocket at the receptor extracellular region, denoted as SBP$_{2-ECL1-1}$, and is formed by the tips of TM1, TM2, and ECL1. Remarkably, in contrast to prior D$_3$R structures – whether in active or inactive conformations – the outermost extracellular residues of TM1 undergo a rearrangement that positions H29$^{1.32}$'s imidazole group, situated between TM2 and TM7, to stack with the 1H-indole group of the ligand SP. Given the absence of H29$^{1.32}$ in preceding D$_3$R cryo-EM[30,31] and crystal structures[25], we sought to ascertain the orientation of the imidazole moiety of H29$^{1.32}$. For this purpose, we performed comparative MD simulations, involving two D$_3$R complexes coupled to Gα$_O$βγ and bound to either FOB02-04A or pramipexole (PDB 7CMU), both within a membrane bilayer and aqueous milieu and executed across five parallel runs of 0.6 μs each. MD

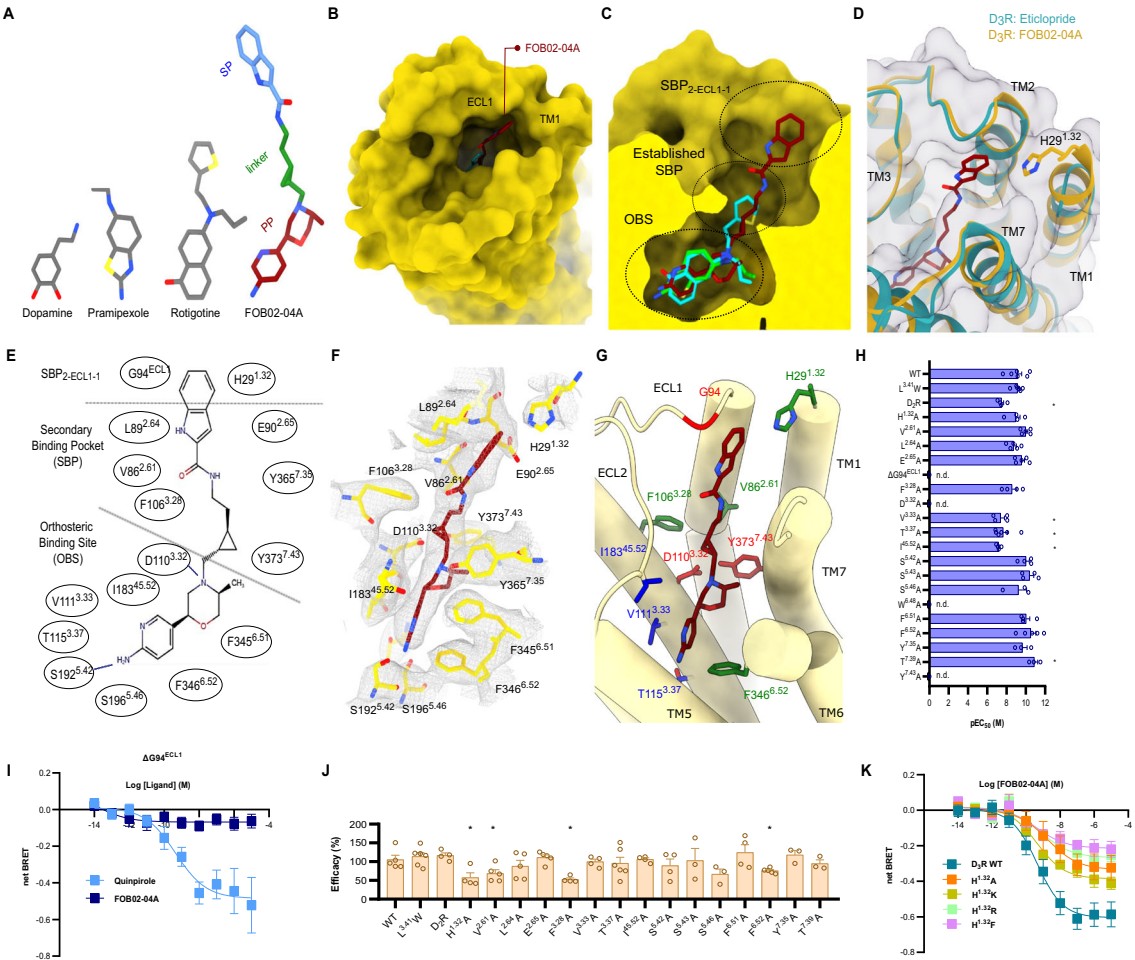

**Fig. 3 | Binding of the bitopic FOB02-04A to the D$_3$R receptor. A** Schematic of dopamine, pramipexole, rotigotine and the bitopic FOB02-04A ligand shown as sticks and colored by component. **B** Binding of the secondary pharmacophore (SP) (sticks, dark red) to a groove-shaped pocket at the D$_3$R (yellow, surface representation) formed by ECL1 and TM1. **C** Two views of a comparison of FOB02-04A (dark red carbon, sticks), pramipexole (green carbon, sticks), and rotigotine (cyan carbon, sticks) binding into the D$_3$R pocket (yellow, surface representation). Dashed circles indicate OBS, established SBP, and the SBP$_{2\text{-ECL1-1}}$ site. **D** Overall binding mode of the bitopic molecule to the D$_3$R and ordering of TM1 upon bitopic binding. FOB02-04A (dark red, sticks) is displayed on superposed structures of D$_3$R bound to eticlopride (cartoon, cyan) and FOB02-04A (cartoon, yellow) **E** Schematic of the FOB02-04A binding into the D$_3$R ligand binding pocket. **F** Binding details of FOB02-04A (dark red, sticks) at the D$_3$R (yellow sticks) with cryo-EM density as gray mesh. **G** Binding details of FOB02-04A (dark red, sticks) at the D$_3$R (yellow cartoons) with residues at the ligand binding pocket colored by functional effect when mutated to alanine: decreased efficacy – green carbons, decreased potency – blue carbon and non-detectable binding – red carbon. **H** pEC$_{50}$ values for alanine mutation of the residues at the ligand binding site in response to G$_{OA}$ activation by FOB02-04A using the TRUPATH assay. All data are means ± SEM of four independent experiments ($n = 4$) performed in technical triplicates except for D110$^{3.32}$A, S196$^{5.46}$A, Y365$^{7.35}$A, T369$^{7.39}$A, W342$^{6.48}$A and Y373$^{7.43}$A for which there was $n = 3$, WT,

V86$^{2.61}$A, L89$^{2.64}$A, E90$^{2.65}$A, ΔG94$^{ECL1}$, F346$^{6.52}$A for there was $n = 5$, and L119$^{3.41}$W, T115$^{3.37}$A for which there was $n = 6$. *$p < 0.05$ (one-way ANOVA with Dunnett post hoc analysis) for D$_2$R ($p = 0.0081$), V111$^{3.33}$A ($p = 0.0049$), T115$^{3.37}$A ($p = 0.0074$) and I183$^{45.82}$A ($p = 0.0013$) and nd - non-detectable. **I** Concentration-response curve of D$_3$R ΔG94$^{ECL1}$ upon G$_{OA}$ activation by quinpirole (light blue, $n = 4$) and FOB02-04A (deep blue, $n = 5$) (shown as net BRET). All data are means ± SEM of the specified biological replicates, each performed in technical triplicates. **J** Emax values for alanine mutation of the residues at the ligand binding site in response to G$_{OA}$ activation by FOB02-04A using the TRUPATH assay. Emax values have been normalized to D$_3$R WT. All data are means ± SEM of four independent experiments performed in technical triplicate ($n = 4$) except for D110$^{3.32}$A, S196$^{5.46}$A, Y365$^{7.35}$A, T369$^{7.39}$A, W342$^{6.48}$A, Y373$^{7.43}$A ($n = 3$), WT, V86$^{2.61}$A, L89$^{2.64}$A, E90$^{2.65}$A, ΔG94$^{ECL1}$, F346$^{6.52}$A ($n = 5$) and L119$^{3.41}$W, T115$^{3.37}$A ($n = 6$). *$p < 0.05$ (Holm-Sidak multiple comparisons tests two-tailed $p$ value) for H29$^{1.32}$A ($p = 0.016$), V86$^{2.61}$A ($p = 0.026$), F106$^{3.28}$A ($p = 0.003$), F346$^{6.52}$A ($p = 0.019$). **K** Concentration-response curves of D$_3$R H29$^{1.32}$A (orange), H29$^{1.32}$F (pink), H29$^{1.32}$K (yellow), and H29$^{1.32}$R (green) upon G$_{OA}$ activation by FOB02-04A (shown as net BRET). All data are means ± SEM derived from three independent experiments ($n = 3$), each performed in technical triplicate except for H29$^{1.32}$A ($n = 4$). All source data within this figure is provided as a Source Data file.

analysis elucidated a more consistent localization of H29$^{1.32}$ between TM2 and TM7 when complexed with FOB02-04A relative to pramipexole. In this conformation, the H29$^{1.32}$ side chain is directed towards the SBP$_{2\text{-ECL1-1}}$, engaging with the SP of FOB02-04A bitopic ligand (Supplementary Fig. 9). Although the protonated N(ε) atom of H29$^{1.32}$ imidazole and the carboxyl entity of E90$^{2.65}$ are too distant to support strong polar or ionic interactions, the D$_3$R complex with the bitopic ligand FOB02-04A exhibited a narrower distance distribution than in pramipexole complex (Supplementary Fig. 9). In addition, in the D$_3$R-

FOB02-04A complex, the N(ε) atom of H29$^{1.32}$ consistently interacts with the backbone oxygen of E90$^{2.65}$. Conversely, when complexed with pramipexole, three of the five trajectories show this distance consistently surpassing 10 Å. This observation reinforces that, while in the FOB02-04A:D$_3$R complex, the H29$^{1.32}$ side chain is predominantly positioned in the SBP$_{2\text{-ECL1-1}}$ where it is stabilized by the ligand, in the pramipexole-bound complex H29$^{1.32}$ side chain points away, likely due to the absence of the allosteric pharmacophore in pramipexole (Supplementary Fig. 9).

To gain further insights into the SBP$_{2\text{-ECL1-1}}$ role, we mutated all residues within this site to alanine (except for G94$^{ECL1}$, which was deleted) and measured ligand-induced activation using BRET2 assays. These experiments revealed that the deletion of G94$^{ECL1}$, which prevents ECL1 from reaching the SP, is essential for FOB02-04A activity (Fig. 3H, I). A previous study identified G94$^{ECL1}$ as a key determinant for binding of a similar bitopic molecule, however, only a reduction in affinity was observed (using radioactive ligands and fluorescence)[45] while, in the current study, ligand-induced activity seemed to be fully ablated. This suggests that FOB02-04A could potentially still bind in the ΔG94$^{ECL1}$ variant (although with lower affinity) but triggers no detectable Gαβγ activation, hence G94$^{ECL1}$ is likely to determine affinity and efficacy. As a control, the ΔG94$^{ECL1}$ variant was activated by quinpirole, a ligand that does not reach ECL1, highlighting the specific effect of the mutation on the activation by the bitopic FOB02-04A (Fig. 3I). Further mutational analysis of residues within SBP$_{2\text{-ECL1-1}}$ identified H29$^{1.32}$ as a key residue, with only a slight reduction in potency (~ 3-fold reduction in EC$_{50}$) but a significant decrease in efficacy upon alanine mutation (Fig. 3J, K). Previous studies predicted how slight variations in the position of the PP at the D$_3$R OBS could modulate compound efficacy[46]. Since several residues at the SBP modulate FOB02-04A efficacy, it is likely that the linker and SP conformation are currently optimal to position the PP for maximal efficacy at the OBS, and that mutations around the SBP restrict conformations of the FOB02-04A PP to less efficacious alternatives. This scheme yields a marked segregation of the functional roles of the protein residues for each bitopic component. While mutations significantly decreasing potency (>100-fold the EC$_{50}$) are primarily found at the OBS, mutations at the SBP mainly decrease FOB02-04A efficacy (Fig. 3G). This suggests that the SP is not only involved in D$_3$R selectivity (see below) but also in optimally positioning the PP for activity. Such conclusions are in line with previous suggestions originating in computational and functional assays[47]. In order to understand the role of H29$^{1.32}$ in SBP binding, we mutated it to residues with different physicochemical properties and sizes (aside from alanine), including H29$^{1.32}$F, H29$^{1.32}$K, and H29$^{1.32}$R. While the H29$^{1.32}$K variant displayed a reduced efficacy similar to H29$^{1.32}$A when activated with FOB02-04A, both H29$^{1.32}$F and H29$^{1.32}$R displayed a further reduction in efficacy, likely due to potential steric clashes with the SBP of the bitopic agonist (Fig. 3K). In contrast, the H29$^{1.32}$A and H29$^{1.32}$F variants did not produce such a marked effect when activated with quinpirole and pramipexole, respectively (Supplementary Fig. 7). Interestingly, the H29$^{1.32}$A variant had reduced efficacy when activated with pramipexole, an agonist that does not reach H29$^{1.32}$ (Supplementary Fig. 7). In the cryo-EM structure as well as in the MD simulation, H29$^{1.32}$ was seen to engage in an H-bond network with E90$^{2.65}$. Such a site might be structurally important for the SBP in the D$_3$R, and hence mutating H29$^{1.32}$ might disrupt the interaction with FOB02-04A but also distort the D$_3$R binding pocket. It is not unusual for residues at the most extracellular sites to have an impact on intrinsic receptor function[48]. Hence, we cannot discard that the functional effects of H29$^{1.32}$ variants on bitopic binding might contain additional contributions not related to ligand binding.

Additional analysis of the MD trajectories with the D$_3$R-FOB02-04A complex suggested a more robust interaction of FOB02-04A with D$_3$R than pramipexole. This was observed by looking at the stable salt bridge interaction between the trans-cyclopropyl amine group of FOB02-04A and the carboxyl group of D110$^{3.32}$ in D$_3$R (which underscores the stable binding pose of the 6-(aminopyridin-3-yl)-5-methyl-morpholine PP moiety) (Supplementary Fig. 6). However, for the pramipexole-bound D$_3$R complex, three out of five MD trajectories displayed substantial deviations in either the equivalent salt bridge interaction with pramipexole amino group, as well as the interactions distance between S196$^{5.46}$ in D$_3$R and the pramipexole's amino group. Since pramipexole and FOB02-04A have similar binding affinities, the propensity of pramipexole towards dissociation observed during MD simulations suggests potential faster association and dissociation rates, in line with the larger bitopic molecule, requiring longer times for association and dissociation (Supplementary Fig. 6).

Overall, the bitopic agonist FOB02-04A uses all three components (PP, linker, and SP) to make critical interactions with the ligand binding pocket since each component contributes with one critical interaction which, if mutated, the ligand-induced activation as a whole, is eliminated. This highlights that selective bitopic molecules are required to bind *en bloc* and that the SP which contains the *address* component is required to contribute significantly to the overall ligand function, otherwise selectivity would be lost.

## Structural basis of FOB02-04A D$_3$R/D$_2$R selectivity

The bitopic FOB02-04A ligand has been designed for its PP to carry the agonist *message* while the SP carries the *address*, and has been reported to be 50-fold more selective for D$_3$R over D$_2$R[9]. Since quantification of selectivity at the D$_3$R/D$_2$R is assay and condition-dependent[6,9,17], we measured the D$_3$R/D$_2$R selectivity using cellular BRET assays, which confirmed the 50-fold selectivity (Supplementary Fig. 7). D$_3$R and D$_2$R have 78% sequence similarity at the transmembrane region and, residues within interacting distance of FOB02-04A, showed high structural similarity and 100% sequence identity at the OBS and established SBP[24]. However, FOB02-04A interactions with the G94$^{ECL1}$ and H29$^{1.32}$ within the SBP$_{2\text{-ECL1-1}}$ form a region that is structurally and sequence-diverse between D$_3$R/D$_2$R. The D$_3$R TM2-ECL1 harbors an extra glycine residue that is absent in D$_2$R (93GGV95 in D$_3$R vs 98GE99 in D$_2$R), which allows this region to interact with the SP in the D$_3$R and not in the D$_2$R (Fig. 4A). Deletion of the extra glycine G94$^{ECL1}$ in D$_3$R ablates ligand-induced activation by FOB02-04A (Fig. 3I and Supplementary Fig. 7). This is in line with previous studies where similar bitopic molecules showed reduced affinity in D$_3$R lacking G94$^{ECL1}$ [45]. This reduction in activity makes G94$^{ECL1}$ the most critical residue for D$_3$R/D$_2$R selectivity. In addition, H29$^{1.32}$ is positioned in TM1, the most sequence-diverse transmembrane helix in GPCRs, and that, within D$_3$R/D$_2$R, shows both sequence and structural diversity (Fig. 4A). Exploiting this unforeseen H29$^{1.32}$ conformation has potential for the development of selective D$_3$R agonists.

## Diversity of the SBP$_{2\text{-ECL1-1}}$ in other aminergic receptors

There are 9 groups of (clinically relevant) closely-related aminergic receptors sub-types (M$_{1-5}$, ARα$_{1A-1D}$, ARα$_{2A-2C}$, ARβ$_{1-3}$, D$_1$, and D$_5$, D$_2$-D$_4$, H$_{3-4}$, 5-HT$_{1A-1F}$, 5-HT$_{2A-2C}$) for which sequence similarity poses problems to generate subtype selective ligands. Selectivity can arise from sequence diversity, structural divergence as well as differences in structural plasticity. Using sequence alignments and the recent explosion in GPCR structural information, we assessed whether the SBP$_{2\text{-ECL1-1}}$ is a novel site of high diversity that could be exploited to develop subtype-selective drugs in other aminergic GPCRs. We first compared the D$_3$R:FOB02-04A complex with structures of other aminergic receptors bound to bitopic ligands that sample different secondary binding sites within the different receptors. Some examples include the D$_2$R bound to spiperone (PDB 7DFP)[49], risperidone (PDB 6CM4)[29] or haloperidol (6LUQ)[50] and the serotonin 5HT$_{1A}$ bound to aripiprazole (7E2Z)[51], IHCH-7179 (8JT6)[52], Vilazodone (8FYL)[53] or buspirone (8FYX)[53]. A structural superposition of these complexes shows that no other ligand interacts with G94$^{ECL1}$ or H29$^{1.32}$ equivalent residues in the D$_2$R or the 5HT$_{1A}$R (Fig. 4B). The closest binding mode would be that of vilazodone binding on the serotonin 5HT$_{1A}$ receptor, where the terminal amide group of vilazodone is close to N100 in the ECL1 (the G94 equivalent), but it is out of H-bonding distance as modeled in the cryo-EM structure.

A further analysis of this site showed that the SBP$_{2\text{-ECL1-1}}$ is variable either in sequence, structure, or both within most aminergic receptor subtypes (Fig. 4C–F). The amount of diversity at the SBP$_{2\text{-ECL1-1}}$ varies within each subfamily, with the least variable being the muscarinic

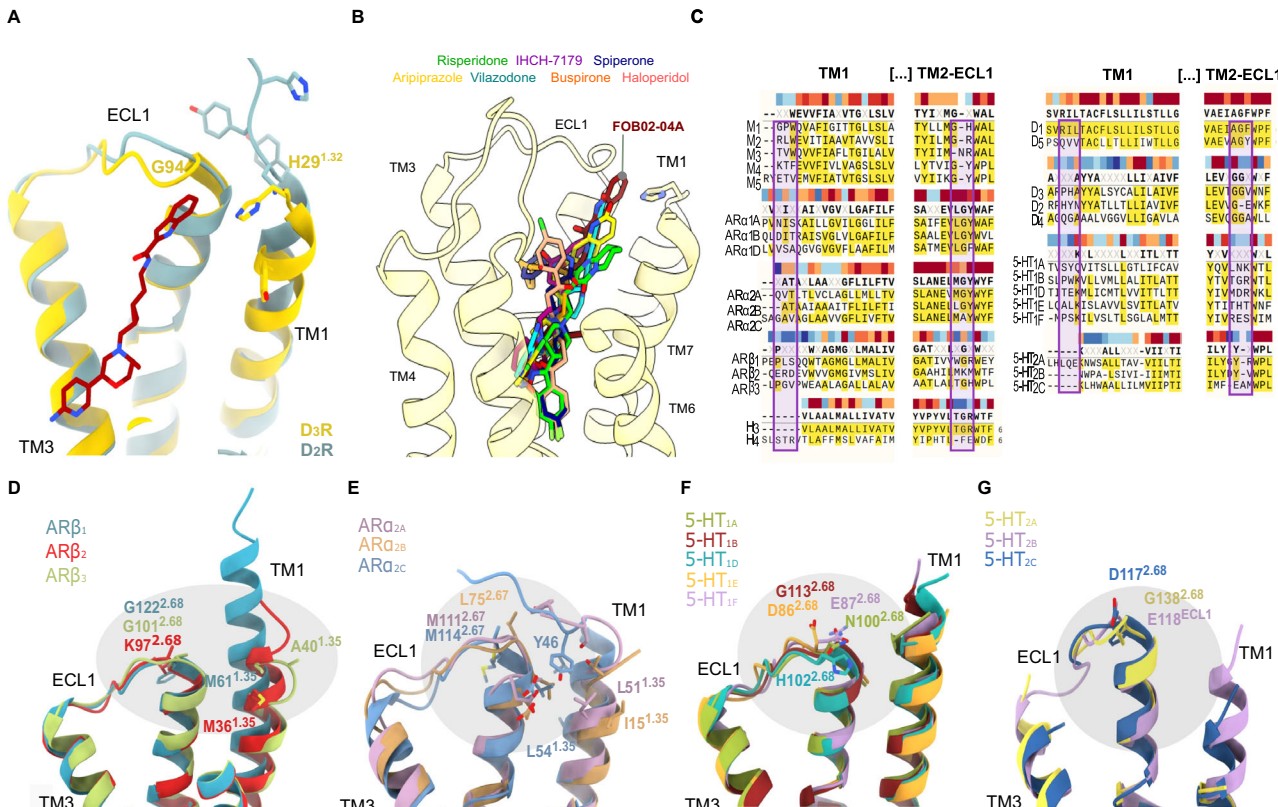

**Fig. 4 | Sequence and structural diversity of the SBP₂-ECL1-1 in aminergic GPCRs.** **A** Comparison of the D₃R (yellow cartoons with relevant residues as sticks) and D₂R (light blue cartoons with relevant residues as sticks) TM2-ECL1 and TM1 regions within reach of FOB02-04A (dark red, sticks). **B** Relative binding sites of other bitopic ligands bound to D₂R (haloperidol (PDB 6LUQ), spiperone (PDB 7DFP), risperidone (PDB 6CM4) and 5-HT₁AR (aripiprazole (PDB 7E2Z), IHCH-7179 (PDB 8JT6), vilazodone (PDB 8FYL) and buspirone (PDB 8FYX) as shown on the

D₃R:FOB02-04A cryo-EM structure. **C** Sequence alignment of TM1 and TM2-ECL1 regions in aminergic GPCRs with residues around the SBP₂-ECL1-1 embedded in a box. Sequence conservation is color-coded above each residue position (gradient from dark red, conserved, to dark blue, non-conserved). Structural differences at the SBP₂-ECL1-1 site among closely related adrenergic receptors (**D**, **E**) and serotonin receptors (**F**, **G**). Receptors are shown as cartoons colored by receptors with relevant residues shown as sticks.

receptors where TM1 is too far apart to contribute in all available structures and the equivalent G94 position is only different in M₃R (N131^ECL1 vs a glycine residue in M₁, M₂, M₄ and M₅). However, there are marked differences in several other subgroups. First, the serotonin 5-HT₁ and 5-HT₂ groups show variable sequence or structure at the G94^ECL1 equivalent position while TM1 is too far apart (Fig. 4C, D). In addition, the recent structural determination of all five dopamine receptors (D₁R-D₅R) highlighted the SBP₂-ECL1-1 as the most variable region between them[30]. Finally, there are groups with marked differences at the SBP₂-ECL1-1 site, e.g., the ARα₂A-₂C subgroup. ARα₂A-₂C shows differences at the G94^ECL1 equivalent position, while they have an increasingly ordered TM1, which could potentially contribute to specific interaction in each receptor. While in ARα₂B TM1 is far apart, it is longer in ARα₂A, where it could contribute with main chain atoms of Y43, and in ADα₂C, where the N-terminus folds over the TM2-ECL1 site providing with additional specific residues (Fig. 4D). In ARβ₁₋₃, the TM2-ECL1 has structural and sequence divergence that could be used to design highly subtype selective bitopic molecules (Fig. 4C). Overall, the SBP₂-ECL1-1 site is a major specificity region that is underexploited for developing subtype-selective drugs. However, this site is far away from the canonical ligand binding site and might be better accessible with bitopic molecules.

**Alternative FOB02-04A conformation at the ligand binding site**
Docking of FOB02-04A to the D₃R reliably reproduced its binding mode when compared to the cryo-EM structure. Yet, a second conformation of FOB02-04A was revealed with comparable docking

scores, suggesting a second plausible orientation (Fig. 5C). In the alternative binding mode, termed Conformation B, the 1H-indole-2-carboxamide SP of FOB02-04A is seen to interact with a less hydrophobic pocket defined by the polar side chains S182^41.51, Y365^7.35, as well as V360^ECL3 and P362^7.32 residues, termed hereafter SBP_ECL2-ECL3. Notably, π-π stacking interactions between the 1H-indole part of FOB02-04A and Y365^7.35 stabilize Conformation B (Fig. 5). MD simulations indicated that the indole SP of FOB02-04A oscillates between SBP₂-ECL1-1 (Conformation A) and the comparatively less hydrophobic SBP_ECL2-ECL3 (Conformation B). A detailed examination of the proximity between D₃R E90^2.65 and the FOB02-04A SP (accentuated with a red palette) juxtaposed with proximity measurements between D₃R Y365^7.35 and the FOB02-04A SP (illustrated in green pallet) provides insights into the temporal predominance of FOB02-04A's Conformation A versus Conformation B (Fig. 5D). Subsequent frequency analyses showed that Conformation A, that engages SBP₂-ECL1-1, is predominant with an estimated 90% prevalence, in contrast to the 10% observed for Conformation B, targeting SBP_ECL2-ECL3 region (Fig. 5D and Supplementary Fig. 6). This information triggered a targeted search for Conformation B within the cryo-EM dataset, which resulted in a model at 3.09 Å resolution (Fig. 1 and Supplementary Figs. 2, 4). In this model, cryo-EM density supports the second conformation for the FOB02-04A SP so as to make π-π stacking interactions with Y365^7.35 in a similar manner as found in docking and MD simulations (Fig. 5A, B and Supplementary Fig. 6). Interestingly, in this cryo-EM map, the extracellular residues of TM1, including H29^1.32, are not resolved, reminiscent of the pramipexole,

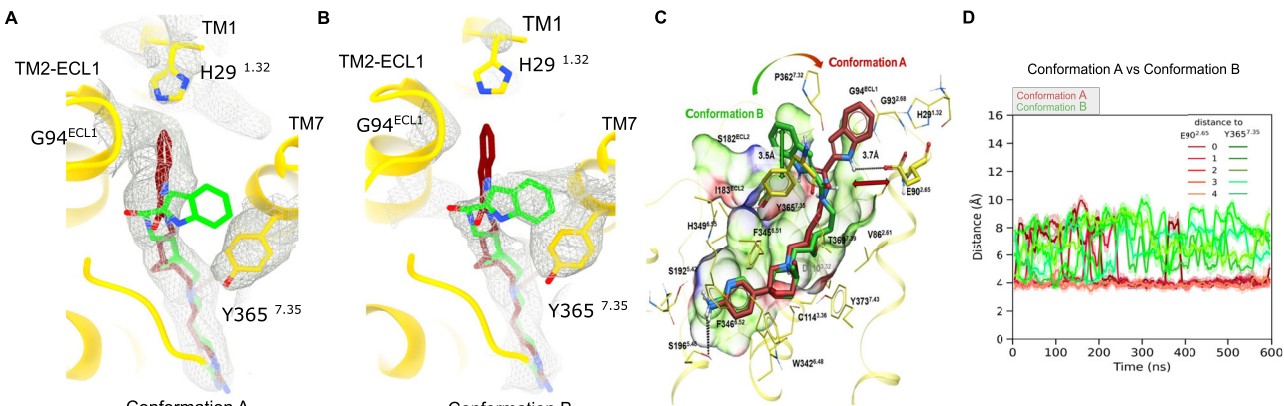

**Fig. 5 | Conformation A and B within the D₃R-G₀:FOB02-04A complex.** Coordinates of the D₃R (yellow cartoon, with relevant residues as sticks) are shown with FOB02-04A in Conformation A (dark red, sticks) superposed to Conformation B (green, sticks). Cryo-EM density is shown as gray mesh for Conformation A (**A**) and Conformation B (**B**) with both superposed FOB02-04A conformations. **C** Predicted binding poses of bitopic FOB02-04A with D₃R obtained by MD simulations showing Conformation A and Conformation B with intramolecular interactions shown as black dashed lines. Black arrows indicate distances for assessing bitopic FOB02-04A binding pose distribution between Conformations A and B with specified closest distances (E90²·⁶⁵ carboxyl group in D₃R to FOB02-04A indole atom N5 and from Y365⁷·³⁵ 4-hydroxyphenyl moiety in D₃R to the phenyl ring of FOB02-04A 1H-indole-2-carboxamide SP). A semi-transparent skin reveals the receptor molecular surface, which is colored by residue properties (red (acidic), blue (basic), green (hydrophobic)). **D** Interaction dynamics between D₃R E90²·⁶⁵ and FOB02-04A SP (depicted in brown palette) compared with proximity distance between D₃R Y365⁷·³⁵ and the SP of FOB02-04A (shown in green palette) suggest that FOB02-04A predominantly adopts Conformation A over B. Data from five independent simulations of the D₃R-Gα₀βγ heterotrimer complex are shown, spanning 0.6 μs of cumulative time per system, with a sampling rate of 10 frames per ns. Solid lines and same-color shadows represent moving average values and one standard deviation, respectively, from 50 frames in all cases.

rotigotine and PD128907 bound D₃R structures (Fig. 5B). This suggests that binding of the SP to the SBP₂₋ₑСₗ₁₋₁ stabilizes the TM1 conformation described above (in agreement with our MD simulations). A comparison of particle numbers between cryo-EM models of Conformation A and B also supported a predominance of Conformation A over B (~60%). In order to understand the role of Conformation B on the function of FOB02-04A, we performed BRET assays on the Y365⁷·³⁵A variant, which exclusively affects Conformation B, resulting in wild-type pharmacological properties (Fig. 3 and Supplementary Fig. 7). This is in stark contrast to the ΔG94ᴱСᴸ¹ variant (only affects Conformation A) which fully ablated Gαβγ dissociation. There are at least two potential explanations for such results. On the first, the bitopic ligand behaves as an antagonist/weak partial agonist when in Conformation B. Such a hypothesis would be in line with our previous observation that residues at the SBP, as well as the position of the SP, are highly relevant for the optimal positioning and efficacy of the PP. In support of this, a minor twist of the PP at the OBS is observed in Conformation B with respect to Conformation A, and minor modifications at the position of the ligand at the D₃R OBS have been shown to regulate ligand efficacy. However, we cannot rule out that the slight difference in PP position is a consequence of the low map resolution. The second explanation would be that Conformation B is an agonist but a very low populated conformation (10% predicted by MD simulations) with too low affinity to be detected in functional assays. This conformation would not be as selective as Conformation A since all residues forming the binding site are conserved between D₂R and D₃R. We assessed whether Conformation B could be occurring at D₂R and could account for the activity of FOB02-04A at D₂R despite not having G94ᴱСᴸ¹ and an H29¹·³² equivalent (with G94ᴱСᴸ¹ being essential for the activity of FOB02-04A in D₃R). However, Y408⁷·³⁵A in D₂R did not result in a signaling loss in functional assays (Supplementary Fig. 7), suggesting that alternative binding modes are likely to exist in D₂R aside from Conformation A and B equivalents. Since D₂R is more plastic than D₃R, the binding mode of this bitopic molecule to D₂R might be hard to predict and structural studies would be required.

## Discussion

Aminergic receptors are highly relevant drug targets, but the high sequence and structural similarity within the family pose a great challenge to developing subtype-selective drugs. Here, we have reported the cryo-EM structure of the human D₃R in complex with the D₃R-selective bitopic agonist, FOB02-04A, and coupled to a G₀ heterotrimer. FOB02-04A binds D₃R with all three components (PP, linker, and SP), fully exploiting the OBS, established SBP, and a new extended SBP₂₋ₑСₗ₁₋₁ that confers FOB02-04A with D₃R selectivity. This SBP₂₋ₑСₗ₁₋₁ is structurally and/or sequence diverse also in aminergic receptors and could potentially be used to develop subtype-selective ligands. Especially interesting is the TM1 contribution to ligand binding since it is the most sequence-diverse transmembrane region in GPCRs, rarely contributes to ligand binding, and could be exploited through the use of bitopic molecules with the required composition and length. Mutational profiling of the ligand binding site showed marked segregation in functional roles of the residues at the OBS and the SBP. While the majority of mutations that impaired potency were located mainly at the OBS, mutations that impaired efficacy were enriched at the SBP. This highlights the relevant role of the SP binding in optimally positioning the PP at the OBS for maximal activity. The computational design of bitopic molecules might benefit from taking such roles into consideration. In addition, the mutational analysis pointed to a mutually PP, linker, and SP-dependent binding mode, i.e., all components contribute with essential interactions for the en bloc binding of the bitopic molecule. This is likely required when higher selectivity is desired since independent binding might yield promiscuous PP binding. Therefore, the message and address components should not be treated as separate entities when developing specific bitopic molecules, but rather working together in tandem with the appropriate linker in between[6].

A second antagonist/non-selective conformation of the FOB02-04A bitopic molecule is proposed, which suggests that care should be taken when developing subtype selective bitopic molecules since the position of the PP at the OBS seems to be altered easily (at least for the D₃R in the case of FOB02-04A) and bitopic molecules tend to be large

and flexible, and alternative non-productive conformations might obscure highly specific and potent conformations in functional assays. Such problems likely contribute to the challenges associated with developing agonist bitopic molecules[6]. Including structural determination in the drug development pipeline is likely to accelerate future progress. Additional structural information on other bitopic-receptor complexes might shed light on this topic.

Regarding the $D_3R/D_2R$ selectivity, a recent report describing the structures of the five dopamine receptors ($D_1R$-$D_5R$) pointed towards $H^{6.55}$ as a specificity determinant, since this residue changes conformation between $D_2$-like receptors in an agonist-dependent manner[30]. While $H^{6.55}$ was located far away from the bitopic molecule under study, molecules with combined interactions at $H^{6.55}$ and the extended $SBP_{2-ECL1-1}$ site have the potential to yield highly specific molecules within $D_2$-like receptors. Such molecules could help to improve current treatments targeting the $D_3R$, a current target for Parkinson's disease and other neurological disorders and neuropsychiatric disorders, including substance use disorders[54–56].

Overall, this work extends the usable SBP in aminergic receptors exploiting an extracellular region of high sequence and structural variability and highlights new insights and pitfalls into the development of highly selective subtype selective bitopic molecules with desired functional efficacies.

## Methods

### Construct design and molecular cloning

All mutagenesis and molecular cloning procedures were performed using the in vivo DNA assembly method[57,58]. The cDNAs of the human $D_3R$ (HASS-FLAG-EGFP-3$C_{protease}$-$D_3R$ with the $L^{3.41}W$ mutation) and human dominant-negative $G\alpha_{OA}$ subunit (S47N$^{G.H1.02}$, G204A$^{G.s3h2.02}$, E246A $^{G.H3.04}$, M249K $^{G.H3.07}$ and A326S $^{G.s6h5.03}$)[24] were obtained through gene synthesis (Gene Fragments, Twist Bioscience) and cloned into the pBacPak8. Rat His$_8$-G$\beta_1$ (pBacPak8), human G$\gamma_2$ (C68S) (pBacPak8), and a baculovirus expressing the scFv16 with a gp67 secretion signal and a C-terminal His$_8$-tag were a gift from Christopher G. Tate's laboratory. For BRET assays in HEK293T cells, the same human HASS-FLAG-EGFP-3$C_{protease}$-$D_3R$ construct was sub-cloned into the pcDNA4/TO vector, upon which all mutants were made (including the wild-type $D_3R$). Constructs containing the $G\alpha_{OA}$-RLuc8, G$\beta_3$, and G$\gamma_9$-GFP2 in pCDNA5 and pCDN3.1 were a gift from Bryan Roth's lab (Addgene plasmid kit # 1000000163). The $G\alpha_{OA}$ mutants to study the $G_{i/O}$ selectivity of the $D_3R$ were done in the $G\alpha_{OA}$-RLuc8 construct. The oligonucleotides used can be found in Supplementary Data 1.

### D$_3$R:Gαβγ:scFv16 production and purification

The scFv16 was produced by infecting *Trichoplusia ni* (Tni, Expression Systems, not authenticated in-house) cells grown in ESF921 media (Expression Systems) at a density of 2-3 × 10⁶ cells/ml and incubated for 48 h at 29 °C. The supernatant was clarified by centrifugation, dialyzed to 20 mM Tris-base pH 8, 500 mM NaCl, and 20 mM imidazole, and loaded onto a pre-equilibrated HisTrap Excel column (Cytiva). The scFv16 was eluted with an imidazole linear gradient, concentrated, and loaded onto a Superdex 200 10/300 GL increase column (Cytiva) equilibrated in 20 mM HEPES pH 7.5 and 100 mM NaCl. Pure protein was concentrated to 4.2 mg/ml, flash-frozen, and stored at – 80 °C until further use.

For the production of the $D_3R$:Gαβγ protein complex, recombinant baculoviruses expressing $D_3R$, $G\alpha_{OA}$, G$\beta_1$ and G$\gamma_2$ were prepared using the FlashBAC ULTRA® system (Oxford Expression Technologies). Tni cells were grown in suspension in ESF921 media to a density of 2-3 × 10⁶ cells/ml, co-infected with $D_3R$, $G\alpha_{OA}$, G$\beta_1$, and G$\gamma_2$ baculoviruses (1:1:1:1 ratio) and shaken at 29 °C for 48 h. Cells were harvested, flash-frozen in liquid nitrogen, and stored at – 80 ˚C for further use. Cell pellets were thawed in 20 mM HEPES pH 7.5, 150 mM NaCl, 10% glycerol, 20 mU/mL apyrase and protease inhibitors cocktail (10 μM E-64,

0.05 μg/ml aprotinin, 0.02 μg/ml leupeptin, 10 μM benzamidine HCl, 0.01 μg/ml pepstatin, 10 μM bestatin and 10 μM PMSF) and incubated with 10 μM of FOB02-04A (compound 53a in ref. [9].) for 30 minutes at 4 °C. Cells were then solubilized with 0.5% (w/v) lauryl maltose neopentyl glycol (LMNG, Anatrace) supplemented with 0.071% (w/v) cholesterol hemisuccinate (CHS, MP Biomedicals ™) at 4 °C for 1 h. The sample was clarified by centrifugation and the supernatant was incubated with Talon Superflow (GE Healthcare Life Sciences) resin overnight at 4 °C. The resin was then washed with 20 column volumes of 20 mM HEPES pH 7.5, 150 mM NaCl, 10% glycerol, 0.007:0.001% LMNG/CHS, 10 μM FOB02-04A and 5 mM imidazole followed by 20 CV of the same buffer with 20 mM imidazole. The sample was eluted with 20 mM HEPES pH 7.5, 150 mM NaCl, 10% glycerol, LMNG 0.003%, 10 μM FOB02-04A and 250 mM imidazole. The complex was concentrated and incubated with pure scFv16 at a molar ratio of 1:1.1 ($D_3R$:Gαβγ:FOB02-04A to scFv16) at 4 °C for 30 min. The resulting complex was purified with a Superose 6 Increase 10/300 GL column (Cytiva) equilibrated with 20 mM HEPES pH 7.5, 25 mM NaCl, 0.003:0.0004% LMNG/CHS and 10 μM FOB02-04A. Pure protein was concentrated at 2.8 mg/ml and FOB02-04A ligand was added to a final concentration of 50 μM.

### Cryo-grid vitrification and data collection

3 μl of $D_3R$:Gαβγ:FOB02-04A:scFv16 at 2.8 mg/ml were applied to 300 mesh Quantifoil 0.6/1 Au grids previously glow discharged with a Leica EM Ace200 Vacuum Coater at 15 mA for 60 s and vitrified with ethane using a Vitrobot Mark IV (FEI Company). Data collection was carried out in a Titan Krios at 300 kV using a K3 detector at the European Synchrotron Radiation Facility (ESRF). A total of 22,655 movies were recorded at a magnification yielding 0.84 Å/pixel with a dose rate of 17.6 e⁻/pixel/s and a defocus range between − 1 to − 3 μm using the Smart EPU Software (ThermoFisher Scientific). Movies were split into 50 frames each and exposed to a total dose of 50 e⁻/Å² (1e⁻/Å² per frame) using a total exposure time of 2 s.

### Cryo-EM Data processing

RELION-4.0[59] was used for all data processing unless otherwise specified. Drift and beam-induced motion correction (5 × 6 patches) were performed using MotionCor2[60] along with dose weighting. Contrast transfer function (CTF) estimation and determination of defocus range were performed with CTFFIND-4.1[61]. Automated particle picking was carried out with Topaz[62]. The initial particles were reduced to 475,951 after 2 rounds of 2D and 3D classifications (using an ab initio model). The best model was refined and subjected to CTF refinement and Bayesian polishing following a 3D classification focused on the receptor (with a mask around the receptor) that yielded 429,908 particles. Refinement of this set of particles yielded a model at 3.16 Å but poor cryo-EM density at the SBP. To improve map quality at the ligand binding site two parallel processing paths were pursued with the 429,908 particle set: 1) a recentering of the particles at the ligand binding site (re-extracted in a 320-pixel box) followed by 3D classification (resulting in 360,038 particles), and 2) 3D classifications with a mask at the extracellular half of the receptor followed by a recentering of the particles at the ligand binding site (as described before) which were further 3D classified (resulting in 176,315 particles). The two sets of particles were merged and duplicates removed, yielding 275,383 particles which were refined using for the last iteration a mask that precluded the $G\alpha_O$-helical domain and the micelle. Post-processing resulted in a cryo-EM map for Conformation A at 3.05 Å. Conformation B was obtained by performing a 3D classification on the 429,908 particles set with a mask on the extracellular half of the receptor, resulting in a model with 252,959 particles, which were subsequently recentered at the ligand binding site and further 3D classified. Particles belonging to the best model, with 201,219 particles, were subjected to heterogeneous refinement and 159,184 particles were lastly refined

through non-uniform refinement in CryoSPARC[63]. This resulted in a cryo-EM map at 3.09 Å according to the gold-standard FSC of 0.143. Local resolution was calculated using CryoSPARC for both models.

## Model building

Model building and refinement were carried out using the CCP-EM software[64] and Phenix[65]. The $D_3R$, $G\beta_1$, $G\gamma_2$, and scFv16 starting coordinates were taken from the $G\alpha_i$-coupled $D_3R$ structure (PDB 7CMV)[31]. The $G\alpha_O$ starting coordinates were taken from the $G\alpha_O$-coupled $\alpha_{2\beta}$ adrenoreceptor structure (PDB 6K41). $D_3R$ was modeled from residue $H29^{1.32}$ to $I223^{5.73}$ and from $R323^{6.29}$ to $C400$ in Conformation A (Conformation B starts at $Y32^{1.35}$). $G\alpha_O$ was modeled from $T4^{G.HN.10}$ to $K54^{G.H1.09}$, $T182^{G.hfs2.05}$ to $V234^{G.s4h3.07}$ and $N242^{G.s4h3.15}$ to $Y354^{G.H5.26}$. Jellybody refinement was performed in REFMAC5[66] followed by manual modification and restraint real space refinement in Coot[67] and Phenix. A dictionary describing the ligand FOB02-04A and its coordinates was created using AceDRG[68] and manually fitted into the density for its latter refinement in real space using Coot and Phenix. B factors were reset to 40 Å² prior to refinement. The final model achieved good geometry (Supplementary Table 1) with validation performed in Coot, EMRinger[69], and Molprobity[70]. The goodness of fit of the model to the map was carried out using Phenix, using a global model-vs-map FSC correlation (Supplementary Table 1).

## Cellular BRET assays

pEC$_{50}$ values were determined using cellular BRET2 assays with the TRUPATH system[26] on HEK293T (ATCC, not authenticated in-house). 50,000 cells/well were seeded in previously poly-lysined 96-well white plates with clear bottoms. The following day, cells were transfected with TransIT-2020 (Mirus Biosciences) at the ratio of 2:1:1:1 of $D_3R$:$G\alpha_{OA}$-RLuc8:$G\beta_3$:$G\gamma_9$-GFP2 (7:1:1:1 for $D_3R$-Y373$^{7.43}$A, $D_3R$-$\Delta$G94$^{ECL1}$ and $D_2R$-Y408$^{7.35}$A) following manufacturer instructions. After 48 h, the medium was replaced by 90 μl/well of freshly prepared assay substrate buffer (1 × Hank's balanced salt solution, 20 mM HEPES pH 7.4, Coelenterazine 400a 7.5 μM). 10 μl of each concentration of FOB02-04A was added and the plate was read using a CLARIOstar (BMG Labtech) with 400 nm (RLuc8-Coelenterazine) and 498.5 nm (GFP2) emission filters at integration times of 1.85 s. BRET ratios were calculated as the ratio of the GFP2 signal to the Rluc8 signal. Equivalent expression of the $G\alpha_{OA}$-RLuc8 variants was confirmed by monitoring luminescence at 400 nm. Data analysis was performed using GraphPad Prism 8.0.1. Data were normalized and a four-parameter logistic curve was fit into the data. Data are presented as mean ± SEM of at least three independent experiments performed in technical triplicate. Source data is provided as a Source Data File.

## Surface expression quantification

HEK293T cells (ATCC, not authenticated in-house) were plated in previously poly-lysine 96-well white plates (50,000 cells/well) and transfected the next day with the $D_3R$ and $D_2R$ variants using PEI MAX® at a 2:1 ratio (PEI:DNA). After 48 h, cells were washed twice with 1X Phosphate Buffered Saline (PBS) and fixed with 4% paraformaldehyde for 20 min at RT. Cells were then washed three times with PBS for 5 min and 100 μl of 1X PBS with 5% BSA (w/v) was added to each well and incubated at RT for 30 min. Subsequently, media was replaced with 1X PBS-5% BSA with an anti-Flag HRP conjugate (1:10,000) and incubated at RT for 30 min. Cells were then rinsed twice with PBS and 50 μl of HRP substrate (Clarity Max™ Western ECL Substrate) was added to each well and incubated for 5 min prior to chemiluminescence detection using a CLARIOstar (BMG Labtech). Data analysis was performed using GraphPad Prism 8.0.1 Chemiluminescence values were normalized to $D_3R$ WT and presented as a ratio of $D_3R$ WT. Data are presented as mean ± SEM of three independent experiments performed in technical triplicate. Source data is provided as a Source Data File.

## Molecular dynamic simulations

The Gromacs simulation engine (version 2020.3)[71] was used to run all molecular dynamics simulations under the Charmm36 force field topologies and parameters[72,73]. Charmm force field parameters and topologies for the ligands FOB02-04A and pramipexole were generated using Charmm-GUI's "Ligand Reader & Modeler" tool[73]. The loop grafting and optimization for modeling missing side chains and loops were performed in the ICM-Pro v3.9-2b molecular modeling and drug discovery suite (Molsoft LLC)[74]. The structurally conserved $H29^{1.32}$, $A30^{1.33}$, and $Y31^{1.34}$ at the N-terminus in the pramipexole bound $D_3R$ (PDB ID: 7CMU)[31] were modeled using human FOB02-04A bound $D_3R$ as the template structure. The lobe in $G\alpha_O$ protein was modeled using a human agonist-bound CB2-$G\alpha_i$ structure (PDB ID: 6PT0)[75]. Structure regularization and torsion profile scanning were done using ICMFF force field[76]. The FOB02-04A-bound and pramipexole-bound structures of $D_3R$ complexes coupled to a $G\alpha_O\beta\gamma$ heterotrimer were then uploaded to the Charmm-GUI webserver[72,77], where the starting membrane coordinates were determined by the PPM[77] server using the Charmm-GUI interface. The complexes were then embedded in a lipid bilayer composed of 1,2-dipalmitoyl-sn-glycero-3-phosphatidylcholine (DPPC), 1,2-dioleoyl-sn-glycero-3-phosphatidylcholine (DOPC), and cholesterol (CHL1) following the recommended ratio of 0.55:0.15:0.30 respectively[78]. The FOB02-04A bound $D_3R$ coupled to a $G\alpha_O\beta\gamma$ heterotrimer contained 220 DPPC, 60 DOPC, and 120 CHL1 lipids, 38818 water molecules, 112 sodium and 104 chloride ions. The pramipexole-bound $D_3R$ coupled to a $G\alpha_O\beta\gamma$ heterotrimer contained 220 DPPC, 60 DOPC, and 120 CHL1 lipids, 37934 water molecules, 108 sodium and 102 chloride ions. Both systems were first subjected to 50000 steps of initial energy minimizations, then 60 ns of equilibration, followed by production runs of up to 600 ns. The simulations were carried out on GPU clusters at the University of Southern California's High-Performance Computing Center. Since the structural insights into the binding mode of the $D_3$ receptor bound to a bitopic agonist were efficiently achieved using standard MD simulations, without the need to explore rare events or surmount significant energy barriers, no enhanced sampling methods were required. The temperature of 310 K and the v-rescale thermostat algorithm were used during the production run[79]. MD simulations were conducted using standard methods without the need for enhanced sampling techniques. The analyses of molecular dynamics trajectories were performed with MDTraj software package[80].

## Molecular docking

The $D_3R$ structure was taken from the current work. The protein-stabilizing single-chain antibody scFv16 was removed from the $D_3R$ structure, leaving the receptor protein subunit. The protein was processed via the addition and optimization of hydrogens and optimization of the side chain residues. Prior to conducting molecular docking, pramipexole underwent chiral definition and formal charge assignment. The compounds' molecular models were created from their two-dimensional representations, and their three-dimensional geometry was refined using the MMFF-94 force field[81]. For docking simulations, a biased probability Monte Carlo (BPMC) optimization approach was employed, adjusting the internal coordinates of the compound based on pre-calculated grid energy potentials of the receptor[82]. The grid potentials, while preserving the receptor's conformational state, considered receptor flexibility through the utilization of "soft" Van der Waals potentials. All-atom docking was performed with the energy-minimized structure of FOB02-04A employing an effort value of 5. The ligand docking box was selected to encompass the extracellular half of the protein for potential grid docking. At least ten independent docking runs were conducted, with the three distinct lowest energy conformations being retained from each run. Consistency across the docking results was assessed by comparing the ligand conformations that achieved the best docking scores. The unbiased docking

procedure did not rely on distance restraints or any predefined information regarding the ligand-receptor interactions. From these docking experiments, two top-scoring docking solutions, referred to as Conformation A and Conformation B, representing FOB02-04A bound to D$_3$R complexes, were further refined. This refinement involved successive rounds of minimization and Monte Carlo sampling, focusing on the ligand conformation and including sidechain residues within 5 Å of the binding site. All the above-mentioned molecular modeling operations were performed in the ICM-Pro v3.9-2b molecular modeling and drug discovery suite (Molsoft LLC)[74].

## Reporting summary
Further information on research design is available in the Nature Portfolio Reporting Summary linked to this article.

## Data availability
The coordinates for the D$_3$R:Gαβγ:scFv16 with the FOB02-04A in conformation A and B have been deposited in the Protein Data Bank with accession codes 9F33 (Conformation A D$_3$R:FOB02-04A structure) and 9F34 (Conformation B D$_3$R:FOB02-04A structure). The cryo-EM maps generated in this study have been deposited in the Electron Microscopy Data Bank (EMDB) with accession codes 50168 (Conformation A D$_3$R:FOB02-04A cryo-EM map) and 50169 (Conformation B D$_3$R:FOB02-04A cryo-EM map). The following existing PDB entries were used in the course of this study: 7CMU, 7CMV, 8IRT, 7DFP, 6CM4, 6LUQ, 7E2Z, 8FYL, 8FYX, 8JT6. The trajectories for the Molecular Dynamics simulations have been deposited as an open-access repo on Zenodo[83]: Nazarova, A. (2024). Molecular Dynamics Trajectories for the D3 receptor (D3R) complexes bound with a Gα$_O$βγ heterotrimer and 1) FOB02-04A bitopic agonist; 2) pramipexole (Zenodo 10800784). Source data are provided in this paper.

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

## Acknowledgements

We thank Christopher Tate´s laboratory for providing the $G_\beta$, $G_\gamma$, and scFv16 baculoviruses. The authors would also like to acknowledge the use of the Servicio General de Apoyo a la Investigación-SAI, Universidad de Zaragoza. We thank Rafael Fernández Fernandez Leiro for providing support with setting up computational facilities for data processing. We thank Reid H. Olsen for their helpful comments while setting up the TRUPATH assays and Robert Nicholls for his help in ligand restraint generation. This work benefited from access to the NeCEN facility and has been supported by iNEXT-Discovery (PID 14410), project number 871037, funded by the Horizon 2020 program of the European Commission and the ESRF (10.15151/ESRF-ES-751565769)(JGN). The authors acknowledge the Center for Advanced Research Computing (CARC) at the University of Southern California for providing computing resources that have contributed to the research results reported within this publication (VK). URL: https://carc.usc.edu. The work was funded by the Ministerio de Ciencia, Innovación y Universidades (PID2020-113359GA-I00), the Spanish Ramón y Cajal program, and the Fondo Europeo de Desarrollo Regional (FEDER) (JGN). A PhD fellowship of the Diputación General de Aragón (SAU). National Institute on Drug Abuse-Intramural Research Program Z1A DA000424 (AHN, AB, FOB).

## Author contributions

J.G.N. and S.A.U. designed the experiments. S.A.U. performed molecular cloning and mutagenesis, receptor expression, purification, complex formation, cryo-EM data collection, cryo-EM data processing, model building, and functional BRET2 assays. A.C.A. performed scFv16 purification and functional BRET2 assays. J.G.N. and S.A.U. performed structure analysis. A.B. and F.O.B. synthesized FOB02-04A, supervised by A.H.N. A.L.N. performed molecular dynamics simulations and docking studies. JHL helped perform molecular dynamics simulations. V.K. supervised the docking studies, molecular dynamics simulations, and manuscript discussion. The manuscript was written mainly by J.G.N. and S.A.U. and included contributions from all authors.

## Competing interests

The authors declare no competing interests.
