## [Peer Review File · Nature Communications]

A bitopic agonist bound to the dopamine 3 receptor reveals a selectivity siteREVIEWER COMMENTS

Reviewer #1 (Remarks to the Author):

This manuscript reports an interesting structural study performed on the dopamine D3 receptor using a bitopic full agonist. The study confirms the already known role of the secondary binding pocket in the selectivity and functional activity of the ligands. The most interesting finding of this study is identifying the TM2-ECL1-TM1 region that might contribute to improve subtype selectivity further. The manuscript reinforces the utility of bitopic approaches on aminergic GPCRs that provided multiple approved drugs. The structural details together with the available structures of aminergic GPCRs with bitopic ligands might help designing new potent and selective compounds with designed functional activity. Although this work applied state of the art methodologies, there are some issues to be clarified before suggesting the manuscript for publication in Nature Comm.

1. The secondary pharmacophore is harboured by the less conserved but not non-conserved secondary pocket. There are many bitopic ligands reported for a number of aminergic GPCRs and there are multiple strategies to design bitopic ligands as exemplified by a number of computational and experimental protocols e.g.

<https://www.ncbi.nlm.nih.gov/pmc/articles/PMC6561289/>

<https://onlinelibrary.wiley.com/doi/full/10.1002/anie.202101478>

<https://pubs.rsc.org/en/content/articlelanding/2021/cc/d1cc04636e>

2. Since there were multiple interactions specific to Gi and Go coupling of the D3 receptor, the G protein selectivity of the receptor should be explained in more details.

3. Figure 3C shows the alignment of monotopic and bitopic D3 agonists and indicates „established SBP”. However, there was no bitopic ligand complexed with D3 receptor earlier and therefore this site cannot be identified as established SBP. This highlights the importance of further comparison of structures with bitopic ligands available for other dopamine and serotonin receptors. In addition, the authors might consider further aminergic receptors with bitopic ligands. Although the authors aligned the sequence of these receptors, this analysis is not enough as structural comparison without comparing the bitopic ligand positions also less than expected. The novelty of the present SBP could only be validated on this way.

4. The role of G94 in the D3 binding of the bitopic ligand cannot be validated by deleting the residue since deletion turns the receptor D2 like. Since the present ligand was claimed to be D3 selective, its binding affinity to the Δ G94 mutant receptor should be demonstrated. DeltaG94 mutation can explain the difference seen in both potency and efficacy if the bitopic compound has no or significantly reduced affinity to this mutant.

5. Since SBP binding was identified as a major factor determining the efficacy, all efficacy values together with SD and statistical significance for the pairs compared should be added.
6. The authors mention several interactions identified during the MD simulations. Facilitating the correct interpretation of these data the full interaction profile obtained from MD simulations should be added to Supplementary. Since conformations A and B were identified during the MD simulations, it would be interesting to disclose their interactions and their relative positions to that of the cryo structures.
7. The authors hypothesize that conformation B is a non-productive partial agonist or antagonist binding mode. This should be investigated by the MD simulation of the approved drug aripiprazole, a D3/D2 partial agonist. The key question is whether aripiprazole binds in A, B or both conformations.

Minor: In the Methods section the authors claimed MD simulations with 60 ns equilibration and up to (?) 500 ns production. However, Fig 5D and other figs in the supplementary show 600 ns simulation time.

Reviewer #2 (Remarks to the Author):

In this study, Arroyo-Urea et al., determined the cryo-EM structure of the D3R coupled to the Go heterotrimer and bound to the D3R selective bitopic agonist FOB02-04A. The FOB02-04A-bound D3R structure reveals its distinct binding mode, which not only occupies the classical orthosteric binding pocket but also protrudes out to contact a new site at the extracellular vestibule of D3R formed by TM2-ECL1 and TM1. Structural analyses combined with mutational studies uncover the activation mechanism of D3R, Go coupling and the major region for subtype selectivity. This study provides new insights into the bitopic structure and allosteric site in aminergic GPCRs, but some concerns should be addressed before its acceptance.

Major comments:

1. The authors mentioned the structural variation in the D3R and G protein binding interface may contribute to G protein selectivity (Go vs Gi). Mutagenesis experiments using BRET are strongly encouraged to support this conclusion.
2. The authors concluded that the ECL1 of D3R harbors an extra glycine residue that is absent in D2R (93GGV95 in D3R vs 98GE99 in D2R) and H29 in TM1 that determine selectivity of FOB02-04A at D2R/D3R. Whether insertion of glycine in D2R or mutation of the equivalent residue in D2R to histidine can enhance the potency of FOB02-04A at D2R?
3. The conclusion that FOB02-04A in the conformational B acts as an antagonist or weak partial agonist requires more evidence. In addition to Y365, the authors can perform more mutagenesis experiments that include residues involved in specific binding with FOB02-04A in the

conformational B. MD simulations using the inactive structure of D3R can be performed to see whether conformational B is more dominant in the inactive state.

Minor comments:

1. MW marker not shown in Supplementary Figure 1A.
2. Emax (Efficacy) of all BRET experiments should be summarized.
3. BW number in Figure 3H should be shown.
4. Cryo-EM flowchart in Supplementary Figure 2 is very confused. It is not clear how the authors can obtain two different conformations. Scale bar should be indicated in Supplementary Figure 2B and 2C.

Reviewer #3 (Remarks to the Author):

The study by Arroyo-Urea et al. reports cryo-EM structures of the dopamine D3 receptor in complex with $G\alpha\beta\gamma$ bound to a bitopic ligand, which adopts two distinct conformations in the ligand-binding pocket. The authors further use signalling assays combined with mutagenesis of residues in the ligand-binding site to gain additional insights into the importance of these residues for signalling in response to FOB02-04A. Molecular dynamics simulations and docking were used to confirm details of the receptor interaction with the G protein and the bitopic ligand.

Overall, this is a technically sound paper employing complementary methods to elucidate the structural details of FOB02-04A binding to the dopamine D3 receptor. The manuscript provides interesting insights into the binding modes of the bitopic ligand and their potential functional differences.

Compared to previously published papers, the authors have solved the structure of dopamine D3 receptor with G_o rather than G_i . They show how the ligand extends beyond the known secondary binding pocket occupied by rotigotine and confirm two binding modes of the secondary pharmacophore using structural and computational methods. Here, it is particularly interesting to note that a second plausible conformation of the ligand was detected in MD simulations and that a search for this conformation in the cryo-EM data set resulted in an additional model. The role of the residues in the binding pocket and the importance of all parts of the bitopic ligand was confirmed by testing $G\alpha_o$ activation of approx. 20 mutants. It is nice to see a thorough exploration of the role of all binding site residues using mutagenesis and signalling assays. The confirmation of the residues involved in binding the secondary pharmacophore will certainly help future studies aiming to find novel ligands for dopamine receptors.

I only have minor comments and suggestions:

1. Have the authors tried to fit a three-parameter model with a slope fixed to 1 to their concentration-response curves? Is there a reason to assume a slope different from 1? As some of the data seem to be on the noisier side of BRET assays using $G\alpha\beta\gamma$ dissociation they may get better estimates of the pEC50. Half-log spacing of ligand concentrations could have helped, too, especially when analysing mutations with a smaller (0.5 - 1 log) pEC50 difference compared to WT.
2. The figure legend of Figure 3 and Supplementary Figure 1/7 mention 'the TRUPATH assay' – There are TRUPATH assays for multiple G proteins, it could be mentioned here (at least in one of the figure legends) that GoB was used.
3. Was the effect of increased or decreased cell-surface expression on potency (pEC50) and efficacy tested? Were the values given in the Table in Supplementary Figure 7 corrected for cell-surface expression? Correction may not be necessary, but it would be nice to see how the signal varies with expression level.
4. I did not find a table listing efficacy values. Those values and their errors would help readers to get a better idea of the magnitude of the changes mentioned in the statements on mutations with decreased efficacy (lines 253, 254, 305).
5. line 200: please define SP and PP abbreviations when they are first used
6. line 310: please consider including the data
7. line 673: could you please give more details on the 3 conformations in each independent docking run?
8. Some authors seem to be missing from the 'Author contributions' section.

Reviewer #4 (Remarks to the Author):

In this article, Arroyo-Urea et al. present the new structure of the D3R in complex with Go, its primary signalling transducer, and the bitopic ligand FOB02-04A. Additional comparative structural analyses, site directed mutagenesis and molecular pharmacology studies, together with molecular dynamics simulations, are used to reveal novel insights into G protein coupling preferences and the GPCR selectivity that could be achieved by targeting an underexploited secondary binding pocket (SBP).

The results presented in this work provide useful details regarding the structural basis of dopaminergic receptor coupling and reveal a new receptor site that can be effectively targeted to achieve selectivity among aminergic GPCRs.

Despite its clear interest, this work would benefit from some additional analyses to fully determine what this new structure can tell us about dopaminergic receptor G protein coupling selectivity, which are the structural determinants of FOB02-04A action on the D3R, and how the detected SBP can provide a source of selectivity in structure-based drug design:

- In page 6, the authors mention previous studies that observed ICL sequences could determine G protein selectivity. However, those articles center their claims on such selectivity mainly stemming from ICL3 and not ICL2, with the latter appearing to be more important in the case of this structure. Based on these same studies (especially Lane et al.), this article would benefit from a comparison of the interactions occurring between this newly solved structure, the structures of D3 in complex with Gi1 and existing structures of D2L, a receptor that couples more promiscuously to Gi proteins, also in complex with Gi1.

- Residue H29(1.32) seems to play an important role in the efficacy of FOB02-04A, but also apparently in the one of pramipexole. Could these changes be related to the fact that mutating this residue to an alanine is preventing the interaction with E90(2.65)? The authors should consider mutations (either virtual, experimental or both) that can help elucidate how this residue is contributing to ligand efficacy (e.g. H29K/R to analyse the importance of a possible bond to TM2 for the stability of the active state of the receptor or H29F if they consider pi stacking could be enough to stabilize FOB02-04A in its agonist binding mode thus promoting receptor activation).

- When it comes to the alternative binding mode of FOB02-04A and its characterisation as an antagonistic / partial agonist conformation, mutation to alanine may also be misleading as the expansion of the binding site resulting from this mutation may still allow the ligand to bind in the secondary conformation. Considering both D1 and D5 receptors have a tryptophan residue in this position, Y7.43W may be a more informative mutation that could be tolerated in D3R in terms of mutagenesis.

- Apart from the addition of generic numbers (please see minor comment below), the analysis in Fig 4 would be much more informative if some quantitative information was added. This could include distances between the analysed helices in different structures or a calculation of the areas corresponding to the newly detected SBP in the different structures.

Minor comments:

- Please add generic numbers to the following figures to facilitate interpretation: Fig 2C and D (CGN), Fig 3H and I (B-W numbering), Fig 4B-F (B-W numbering), Fig S7 (B-W numbering).

- Some claims are difficult to interpret or should be rephrased to avoid misinterpretations:

a) Page 3: 'D2R and D3R differ in brain distribution and signaling properties, and are both targeted by current antipsychotics and drugs for the treatment of neurological diseases (such as Parkinson's disease 15,16). Although agonists with some selectivity exist, treatments are still suboptimal due to a lack of selectivity and different effects originating from the activation of both receptors 17,18.' Especially in the case of antipsychotics, multi-target binding affinities have been postulated as a requirement for drug efficacy (see Roth, Nat Rev Drug Discov 2004) and in drugs like cariprazine, D2/D3 effects are believed to be required for antipsychotic action. Therefore this statement could be misleading and should be reformulated.

b) Page 4: please recheck 'allowing to separate two FOB02-04A and binding site conformations'.

c) Page 5: please recheck 'No major conformational changes are found in the D3R when comparing it bound to pramipexole'.

d) Page 7: 'The cryo-EM density allowed modelling of the three bitopic components.' Does this mean 'The cryo-EM density allowed modelling of the three bitopic moieties.' or maybe 'The cryo-EM density allowed modelling of the three components of the bitopic ligand.'

e) Page 8: 'This suggests a direct role in ligand-induced activation of the bitopic molecule.' Do the authors mean that this suggests a direct role of the bitopic molecule in ligand-induced receptor activation? In this sense, some of the mutations suggested above could provide further evidence to substantiate this claim.

Point-by-point response to reviewers (responses in blue)

We thank the reviewers for their comments. We found them very constructive, and we honestly believe they have contributed to improve the manuscript. We provide below a point-by-point answer to each of their comments:

Reviewer #1 (Remarks to the Author):

This manuscript reports an interesting structural study performed on the dopamine D3 receptor using a bitopic full agonist. The study confirms the already known role of the secondary binding pocket in the selectivity and functional activity of the ligands. The most interesting finding of this study is identifying the TM2-ECL1-TM1 region that might contribute to improve subtype selectivity further. The manuscript reinforces the utility of bitopic approaches on aminergic GPCRs that provided multiple approved drugs. The structural details together with the available structures of aminergic GPCRs with bitopic ligands might help designing new potent and selective compounds with designed functional activity. Although this work applied state of the art methodologies, there are some issues to be clarified before suggesting the manuscript for publication in Nature Comm.

1. *The secondary pharmacophore is harboured by the less conserved but not non-conserved secondary pocket. There are many bitopic ligands reported for a number of aminergic GPCRs and there are multiple strategies to design bitopic ligands as exemplified by a number of computational and experimental protocols e.g.*

<https://www.ncbi.nlm.nih.gov/pmc/articles/PMC6561289/>

<https://onlinelibrary.wiley.com/doi/full/10.1002/anie.202101478>

<https://pubs.rsc.org/en/content/articlelanding/2021/cc/d1cc04636e>

We thank the reviewer for pointing us to this literature. We found it useful and have now included it within the manuscript together with a statement about the availability of computational and chemical strategies to develop bitopic compounds (line 92).

2. *Since there were multiple interactions specific to G_i and G_o coupling of the D3 receptor, the G protein selectivity of the receptor should be explained in more details.*

We agree with the reviewer that obtaining insights into the G protein coupling selectivity of D₃R would be of relevance. For this purpose, we have measured D₃R G_i potency and efficacy upon agonist treatment with the bitopic FOB02-04A, confirming the lower preference of D₃R for G_i . We have then identified all residues involved at the D₃R: $G\alpha$ protein interface which differed between G_o and G_i and we reverted them one at a time to G_i over the G_o background. We then performed functional assays in HEK293T cells using BRET2 as previously done in this manuscript. Overall, all mutations had an impact on the G_o protein coupling with residue G350^{G.H5.22} located at the C-terminal $\alpha 5$ having the most impact and hence the primary determinant of D₃R: G_o selectivity. We found that G350^{G.H5.22} from G_o packs tightly against ICL2, which in $G\alpha_i$ the bulkier G350D^{G.H5.22} substitution might present steric strains, hindering G_i coupling. MD simulations support the sampling of H140^{34.55} to occupy the gap left at the G350^{G.H5.22} which would not be possible with G_i . We also found that in D₂R (a receptor without $G_{i/o}$ selectivity), ICL2 is displaced outwards from the receptor core, yielding a wider cavity and posing no steric restrictions to either $G\alpha_o$ or $G\alpha_i$ coupling at this position.

We have included these findings in the main text (“Activation mechanism and G_o coupling of the D₃R bound to FOB02-04A” section), have added the data to Fig 2 and Supplementary Fig 5 and included the additional experiments in Methods.

3. Figure 3C shows the alignment of monotopic and bitopic D3 agonists and indicates „established SBP”. However, there was no bitopic ligand complexed with D3 receptor earlier and therefore this site cannot be identified as established SBP. This highlights the importance of further

comparison of structures with bitopic ligands available for other dopamine and serotonin receptors. In addition, the authors might consider further aminergic receptors with bitopic ligands. Although the authors aligned the sequence of these receptors, this analysis is not enough as structural comparison without comparing the bitopic ligand positions also less than expected. The novelty of the present SBP could only be validated on this way.

We apologize for the confusion. We have termed established SBP to the pocket and residues that had been previously used to define the SBP in aminergic receptors as described in Michino et al, 2015 (PMID 25527701) and Eyged et al, 2021 (PMID 33826962), and that have been widely used in the design of ligands using computation and medicinal chemistry. We have termed established SBP to such site to differentiate it from the new site which the bitopic ligand in the current manuscript describes, which extends the usable pocket to generate selective molecules. To our knowledge, there are no experimental structures of bitopic ligands for other dopamine or serotonin receptors (at least that have proven to be bitopic, i.e. individual components can bind on their own), hence we cannot perform a structural comparison to such structures.

4. The role of G94 in the D3 binding of the bitopic ligand cannot be validated by deleting the residue since deletion turns the receptor D2 like. Since the present ligand was claimed to be D3 selective, its binding affinity to the Δ G94 mutant receptor should be demonstrated. DeltaG94 mutation can explain the difference seen in both potency and efficacy if the bitopic compound has no or significantly reduced affinity to this mutant.

It is indeed difficult to establish the role of the G94 since its mutation or deletion likely forces the ligand to transition into conformation B. We have attempted to perform experiments to measure the IC_{50} of the FOB02-04A bitopic ligand, assuming that the Δ G94 variant (likely in conformation B), however the results have been inconclusive due to technical challenges when working with the Δ G94 variant. In any case, we are aiming to obtain functionally selective agonists and hence we feel functionality and not binding affinity, better represents whether the compound is selective or not. In the case of FOB02-04A, removal of G94 renders this bitopic agonist fully inactive. Furthermore, we have made additional controls to include the activation of the Δ G94 variant by other agonists that do not interact with this region, validating the functionality of the construct (Fig 2 and Supplementary Fig 7).

5. Since SBP binding was identified as a major factor determining the efficacy, all efficacy values together with SD and statistical significance for the pairs compared should be added.

We thank the reviewer for this suggestion, this suggestion was of great value. We have now calculated E_{max} for all experiments and have added them to the table in Supp Fig 7 as well as adding them in Fig 3.

6. The authors mention several interactions identified during the MD simulations. Facilitating the correct interpretation of these data the full interaction profile obtained from MD simulations should be added to Supplementary. Since conformations A and B were identified during the MD simulations, it would be interesting to disclose their interactions and their relative positions to that of the cryo structures.

This is a great suggestion. We have now added a 2D full interaction profile of the bitopic FOB02-04A in conformation A and B from MD simulations to Supplementary Figure 6. Additionally, we have added two panels that show a comparison of conformations A and B in MDs, cryo-EM structures and docking studies in Supplementary Figure 6.

7. The authors hypothesize that conformation B is a non-productive partial agonist or antagonist binding mode. This should be investigated by the MD simulation of the approved drug

aripiprazole, a D3/D2 partial agonist. The key question is whether aripiprazole binds in A, B or both conformations.

Although an interesting suggestion, aripiprazole is a different molecule to the bitopic ligand investigated in this work. Whether aripiprazole can bind in two conformations and whether they have different functionalities would not confirm or exclude the fact that it could occur in the current bitopic ligand. In turn, we hope this work helps understand aripiprazole binding and functionality in the future. Instead, during this revision we have focused our efforts in trying to understand whether conformation B of the bitopic ligand is really an antagonist although with limited success (see reply to reviewer 2). We have therefore opted for discussing the different possibilities that would explain the functional data, which would include that the conformation B in an antagonist/weak agonist or it is a very low populated conformation with low affinity that is challenging to detect on its own (end of “Alternative FOB02-04A conformation at the ligand binding site” section, lines 443-468).

Minor: In the Methods section the authors claimed MD simulations with 60 ns equilibration and up to (?) 500 ns production. However, Fig 5D and other figs in the supplementary show 600 ns simulation time.

Thank you, this was a mistake and indeed simulations were 600ns, we have now corrected it.

Reviewer #2 (Remarks to the Author):

In this study, Arroyo-Urea et al., determined the cryo-EM structure of the D3R coupled to the Go heterotrimer and bound to the D3R selective bitopic agonist FOB02-04A. The FOB02-04A-bound D3R structure reveals its distinct binding mode, which not only occupies the classical orthosteric binding pocket but also protrudes out to contact a new site at the extracellular vestibule of D3R formed by TM2-ECL1 and TM1. Structural analyses combined with mutational studies uncover the activation mechanism of D3R, Go coupling and the major region for subtype selectivity. This study provides new insights into the bitopic structure and allosteric site in aminergic GPCRs, but some concerns should be addressed before its acceptance. Major comments:

1. The authors mentioned the structural variation in the D3R and G protein binding interface may contribute to G protein selectivity (Go vs Gi). Mutagenesis experiments using BRET are strongly encouraged to support this conclusion.

We agree with the reviewer that obtaining insights into the G protein coupling selectivity of D3R would be of relevance. For this purpose, we have measured D3R Gi potency and efficacy upon agonist treatment with the bitopic FOB02-04A, confirming the lower preference of D3R for Gi. We have then identified all residues involved at the D3R:Gα protein interface which differed between Go and Gi and we reverted them one at a time to Gi over the Go background. We perform functional assays in HEK293T cells using BRET2 as previously done in this manuscript. Overall, all mutations had an impact on the Go protein coupling with residue G350^{G.H5.22} located at the C-terminal α5 having the most impact and hence the primary determinant of D3R:Go selectivity. We found that G350^{G.H5.22} from Go packs tightly against ICL2, which in Ga_i the bulkier G350D^{G.H5.22} substitution might present steric strains, hindering Gi coupling. MD simulations support the sampling of H140^{34.55} to occupy the gap left at the G350^{G.H5.22} which would not be possible with Gi. We also found that in D2R (a receptor without Gi/Go selectivity), ICL2 is displaced outward from the receptor core, yielding a wider cavity and posing no steric restrictions to either Gα_o or Gα_i coupling at this position (Fig 2C).

We have included these findings in the main text (“Activation mechanism and G_o coupling of the D₃R bound to FOB02-04A” section), have added the data to Fig 2 and Sup Fig 5 and included the additional experiments in Methods.

2. The authors concluded that the ECL1 of D3R harbors an extra glycine residue that is absent in D2R (93GGV95 in D3R vs 98GE99 in D2R) and H29 in TM1 that determine selectivity of FOB02-04A at D2R/D3R. Whether insertion of glycine in D2R or mutation of the equivalent residue in D2R to histidine can enhance the potency of FOB02-04A at D2R?

This is an interesting suggestion; however, we initially discarded it due to the fact that, based on mutational data at the D₃R and the D₂R, the bitopic ligand is binding with a totally different mode at the D₂R than at the D₃R, different from Conformation A or B. This is due to the enhanced plasticity of the D₂R vs the D₃R, also proposed in previous publications, and the flexibility of the bitopic ligand. We believe this is one of the reasons why it has been so challenging to develop selective D₃R full agonists. Therefore, mutations at the D₂R to mimic the SBP of Conformation A will not be very telling and understanding how the bitopic ligand binds D₂R would require structural determination (which would be outside the scope of this work).

3. The conclusion that FOB02-04A in the conformational B acts as an antagonist or weak partial agonist requires more evidence. In addition to Y365, the authors can perform more mutagenesis experiments that include residues involved in specific binding with FOB02-04A in the conformational B. MD simulations using the inactive structure of D3R can be performed to see whether conformational B is more dominant in the inactive state.

During this revision we have put a great amount of effort to understand whether conformation B of the bitopic ligand is really an antagonist. Regarding the use of additional mutants to detect conformation B, it is indeed an interesting idea, unfortunately Y365 is the only residue that is specific to conformation B, and hence we could not further extend the study to additional mutations.

To understand whether conformation B is an antagonist we have:

- a) Performed competition assays using BRET cellular assays where we used the ΔG94 variant, aiming to force bitopic binding into conformation B, and then aimed to compete quinpirole or titrated quinpirole in the presence of the bitopic molecule FOB02-04A.
- b) Used fluorescently labeled dopamine to perform similar competition assays on the ΔG94 D₃R variant expressed in HEK293T cells.

Unfortunately, working with the ΔG94 variant is technically challenging and we did not obtain conclusive results in either direction when performing either type of experiments (a or b).

- c) We have performed MD simulations to understand whether conformation B was stabilizing the inactive conformation more than conformation A. For this purpose, both conformations were docked into the crystal structure of the inactive-state D₃R and subject to MD simulations (see Figure 1 below). This experiment did not show clear trends; however, this strategy would only show results if conformation B were to be an inverse agonist and will not show a trend if the bitopic ligand is a very weak agonist/antagonist.

Overall, despite significant and multiple efforts, we have not been able to further support the idea (or reject it) that conformation B is an antagonist, further than what we can already do with the current data. We have therefore expanded that section and have described the different scenarios that would result in a loss of activity upon Gly94 removal, one option being the loss of efficacy of the ligand in conformation B, and another option being that conformation B is a low populated conformation with poor affinity and hence not detected. We cannot discard either that our efforts

to force conformation A into conformation B by deleting Gly94 are indeed not resulting in what we desire.

Figure 1. Distribution of binding poses of bitopic FOB02-04A in D3R-inactive-state complex. (A) Predicted binding poses of bitopic FOB02-04A with inactive-state dopamine D3 receptor (D₃R). D₃R is shown in cyan, the docked conformation A (red) and conformation B (green) of FOB02-04A are shown as capped sticks, and intramolecular interactions are shown as black dashed lines. Within the binding pocket, residues interacting with conformations A and B are depicted as sticks, with E90^{2.65} and Y365^{7.35} highlighted in bold. Arrows indicate distances for assessing bitopic FOB02-04A's binding pose distribution between conformations A and B, specifically from E90^{2.65}'s carboxyl group in D₃R to FOB02-04A's indole atom N5 (red) and from Y365^{7.35}'s 4-hydroxyphenyl moiety in D₃R to the phenyl ring of FOB02-04A's 1H-indole-2-carboxamide SP (green), the closest distances are depicted as well; (B) Hydrogen bond interaction dynamics between D3R's E90^{2.65} carboxyl group and FOB02-04A's indole atom N5 (depicted in brown pallet) compared with proximity distance between 4-hydroxyphenyl moiety of D3R's Y365^{7.35} and phenyl ring of 1H-indole-2-carboxamide SP of FOB02-04A's (shown in green palette) for **ConformationA-bound D3R**; (C) Hydrogen bond interaction dynamics between D3R's E90^{2.65} carboxyl group and FOB02-04A's indole atom N5 (depicted in brown pallet) compared with proximity distance between 4-hydroxyphenyl moiety of D3R's Y365^{7.35} and phenyl ring of 1H-indole-2-carboxamide SP of FOB02-04A's (shown in green palette) for **ConformationB-bound D3R**.

Minor comments:

1. MW marker not shown in Supplementary Figure 1A.

Thank you, we have now added the MW marker.

2. E_{max} (Efficacy) of all BRET experiments should be summarized.

We have calculated E_{max}, their errors and statistical significance for all experiments and have added them to the table in Supp Fig 7 as well as Fig 3.

3. BW number in Figure 3H should be shown.

We have now included BW numbering to all figures.

4. Cryo-EM flowchart in Supplementary Figure 2 is very confused. It is not clear how the authors can obtain two different conformations. Scale bar should be indicated in Supplementary Figure 2B and 2C.

We understand the cryo-EM flowchart is complicated; disentangling a second conformation of the bitopic ligand has been a challenge. We have modified it to enhance clarity. We have also added a scale bar to Supplementary Fig 2B and 2C.

Reviewer #3 (Remarks to the Author):

The study by Arroyo-Urea *et al.* reports cryo-EM structures of the dopamine D3 receptor in complex with Ga α 3 β bound to a bitopic ligand, which adopts two distinct conformations in the ligand-binding pocket. The authors further use signalling assays combined with mutagenesis of

residues in the ligand-binding site to gain additional insights into the importance of these residues for signalling in response to FOB02-04A. Molecular dynamics simulations and docking were used to confirm details of the receptor interaction with the G protein and the bitopic ligand.

Overall, this is a technically sound paper employing complementary methods to elucidate the structural details of FOB02-04A binding to the dopamine D3 receptor. The manuscript provides interesting insights into the binding modes of the bitopic ligand and their potential functional differences.

Compared to previously published papers, the authors have solved the structure of dopamine D3 receptor with Go rather than Gi. They show how the ligand extends beyond the known secondary binding pocket occupied by rotigotine and confirm two binding modes of the secondary pharmacophore using structural and computational methods. Here, it is particularly interesting to note that a second plausible conformation of the ligand was detected in MD simulations and that a search for this conformation in the cryo-EM data set resulted in an additional model. The role of the residues in the binding pocket and the importance of all parts of the bitopic ligand was confirmed by testing GaoB activation of approx. 20 mutants. It is nice to see a thorough exploration of the role of all binding site residues using mutagenesis and signalling assays. The confirmation of the residues involved in binding the secondary pharmacophore will certainly help future studies aiming to find novel ligands for dopamine receptors.

I only have minor comments and suggestions:

1. Have the authors tried to fit a three-parameter model with a slope fixed to 1 to their concentration-response curves? Is there a reason to assume a slope different from 1? As some of the data seem to be on the noisier side of BRET assays using $G\alpha\beta\gamma$ dissociation they may get better estimates of the pEC50. Half-log spacing of ligand concentrations could have helped, too, especially when analysing mutations with a smaller (0.5 - 1 log) pEC50 difference compared to WT.

We thank the reviewer for his/her suggestion, we initially aimed for a 4-parameter curve to capture any unexpected behaviour, however, at this point we indeed do not expect a slope different from 1. We have tested a fit with a fixed slope of 1 in our data and the results have remained virtually identical, hence we have opted to leave the current analysis. We have also performed additional curves aiming to reduce the errors in some of the plots.

2. The figure legend of Figure 3 and Supplementary Figure 1/7 mention ‘the TRUPATH assay’ – There are TRUPATH assays for multiple G proteins, it could be mentioned here (at least in one of the figure legends) that GoB was used.

We have now added that we use G_0 in BRET experiments in all figures containing BRET assays (Fig 3, S1 and S7) as well as in Methods.

3. Was the effect of increased or decreased cell-surface expression on potency (pEC50) and efficacy tested? Were the values given in the Table in Supplementary Figure 7 corrected for cell-surface expression? Correction may not be necessary, but it would be nice to see how the signal varies with expression level.

Indeed, expression levels require attention and we have aimed to have a similar expression of D₃R variants in the assays by quantifying surface expression using chemiluminescence and compensating with additional transfected DNA amount where necessary. Some discrepancies in surface expression are somehow unavoidable. For this reason, we initially tested different amounts of transfected DNA for the receptor and do not see an effect in efficacy. Additionally, TRUPATH BRET assays that monitor G protein activation have been shown to be independent from expression levels, due to the avoidance of signal amplification within the cell (Olsen, 2021,

Nat Chem Biol). Hence current efficacy values are independent of expression variability within the range found in this work and do not require normalization.

4. I did not find a table listing efficacy values. Those values and their errors would help readers to get a better idea of the magnitude of the changes mentioned in the statements on mutations with decreased efficacy (lines 253, 254, 305).

We have calculated Emax, their errors and statistical significance for all experiments and have added them to the table in Supp Fig 7 as well as Fig 3.

5. line 200: please define SP and PP abbreviations when they are first used

We have defined SP and PP abbreviations in the first half of the introduction (line 81-82).

6. line 310: please consider including the data.

We have now added the pharmacological characterization of the H29A when activated by pramipexol in Supplementary Figure 7. We have also mutated H29 to arginine, phenylalanine and lysine and measure the pharmacological profiles (in response to another reviewer). These are now shown in Fig 3.

7. line 673: could you please give more details on the 3 conformations in each independent docking run?

This is technical information from the docking algorithm, we have now updated the methods section to explain this clearly.

8. Some authors seem to be missing from the 'Author contributions' section.

Thank you, we have now added the missing contributor.

Reviewer #4 (Remarks to the Author):

In this article, Arroyo-Urea et al. present the new structure of the D3R in complex with Go, its primary signalling transducer, and the bitopic ligand FOB02-04A. Additional comparative structural analyses, site directed mutagenesis and molecular pharmacology studies, together with molecular dynamics simulations, are used to reveal novel insights into G protein coupling preferences and the GPCR selectivity that could be achieved by targeting an underexploited secondary binding pocket (SBP).

The results presented in this work provide useful details regarding the structural basis of dopaminergic receptor coupling and reveal a new receptor site that can be effectively targeted to achieve selectivity among aminergic GPCRs.

Despite its clear interest, this work would benefit from some additional analyses to fully determine what this new structure can tell us about dopaminergic receptor G protein coupling selectivity, which are the structural determinants of FOB02-04A action on the D3R, and how the detected SBP can provide a source of selectivity in structure-based drug design:

- In page 6, the authors mention previous studies that observed ICL sequences could determine G protein selectivity. However, those articles center their claims on such selectivity mainly stemming from ICL3 and not ICL2, with the latter appearing to be more important in the case of this structure. Based on these same studies (especially Lane et al.), this article would benefit from

a comparison of the interactions occurring between this newly solved structure, the structures of D3 in complex with Gi1 and existing structures of D2L, a receptor that couples more promiscuously to Gi proteins, also in complex with Gi1.

We agree with the reviewer that obtaining insights into the G protein coupling selectivity of D₃R would be of relevance. For this purpose, we have measured D₃R G_i potency and efficacy upon agonist treatment with the bitopic FOB02-04A, confirming the lower preference of D₃R for G_i. We have then identified all residues involved at the D₃R:G α protein interface which differed between G_O and G_i and we reverted them one at a time to G_i over the G_O background. We then performed functional assays in HEK293T cells using BRET2 as previously done in this manuscript. Overall, all mutations had an impact on the G_O protein coupling with residue G350^{G.H5.22} located at the C-terminal $\alpha 5$ having the most impact and hence the primary determinant of D₃R:G_O selectivity. We found that G350^{G.H5.22} from G_O packs tightly against ICL2, which in G α_i the bulkier G350D^{G.H5.22} substitution might present steric strains, hindering G_i coupling. MD simulations support the sampling of H140^{34.55} to occupy the gap left at the G350^{G.H5.22} which would not be possible with G_i. We also found that in D₂R (a receptor without G_{i/O} selectivity), ICL2 is displaced outwards from the receptor core, yielding a wider cavity and posing no steric restrictions to either G α_O or G α_i coupling at this position.

We have included these findings in the main text (“Activation mechanism and G_O coupling of the D₃R bound to FOB02-04A” section), have added the data to Fig 2 and Sup Fig 5 and included the additional experiments in Methods.

- Residue H29(1.32) seems to play an important role in the efficacy of FOB02-04A, but also apparently in the one of pramipexole. Could these changes be related to the fact that mutating this residue to an alanine is preventing the interaction with E90(2.65)? The authors should consider mutations (either virtual, experimental or both) that can help elucidate how this residue is contributing to ligand efficacy (e.g. H29K/R to analyse the importance of a possible bond to TM2 for the stability of the active state of the receptor or H29F if they consider pi stacking could be enough to stabilize FOB02-04A in its agonist binding mode thus promoting receptor activation).

This is a very interesting point. We have now mutated H29^{1.32} to arginine, phenylalanine and lysine. Mutations to lysine yields a reduced efficacy, similar to the H29^{1.32}A mutation, while the H29^{1.32}R and H29^{1.32}F variants further reduce the efficacy. This is potentially due to steric issues with the indole group of the bitopic agonist. The interaction between H29^{1.32} and E90^{2.65} might indeed be of relevance in maintaining the geometry of the pocket somehow. For all this, we have now added the experimental data (shown in Fig 3 and Supplementary Fig 7) as well as a discussion of the mutants and the potential role of the interaction between H29^{1.32} and E90^{2.65} to the manuscript (lines 326 to 340).

- When it comes to the alternative binding mode of FOB02-04A and its characterization as an antagonistic / partial agonist conformation, mutation to alanine may also be misleading as the expansion of the binding site resulting from this mutation may still allow the ligand to bind in the secondary conformation. Considering both D1 and D5 receptors have a tryptophan residue in this position, Y7.43W may be a more informative mutation that could be tolerated in D3R in terms of mutagenesis.

From docking and MD simulation studies we can confirm that the indole group makes a strong stacking interaction with Y365, hence mutation to alanine should remove such interaction, or in the worst scenario, have some functional impact. Although additional mutations might further confirm the conclusions, the full lack of impact of this mutation suggest that the bitopic agonist binds in a totally different manner at the D₂R, which is supported by the higher flexibility of the receptor. At this point we believe that only a full structural determination of D₂R in complex with

FOB02-04A would provide the necessary information (which unfortunately would be outside the scope of this work).

- Apart from the addition of generic numbers (please see minor comment below), the analysis in Fig 4 would be much more informative if some quantitative information was added. This could include distances between the analysed helices in different structures or a calculation of the areas corresponding to the newly detected SBP in the different structures.

We found this was a great idea and have analyzed the SBP in different ways, calculating surface areas as well as inter-helical distances to quantify pocket properties as well as to assess whether TM1 could readily contribute to ligand binding if a bitopic molecule was to position in a similar manner as in the D₃R. However, we found it challenging for the following reasons: a) there is no defined and/or isolated pocket at the G94/H29 position, but it is rather an extension of the usable pocket within the D₃R, b) although inter-helices distances might seem like a good idea, they end-up not being representative of whether TM1 would contribute or not, since it is more dependent on the side chain size, and c) as occurs with the D₃R, a reorganization triggered the contribution of TM1 to ligand binding, hence potential reorganizations within the flexible TM1 are unpredictable. In order to ensure clarity, we have now termed “site” rather than “pocket” when we refer exclusively to the region formed by the new residues.

Minor comments:

- Please add generic numbers to the following figures to facilitate interpretation: Fig 2C and D (CGN), Fig 3H and I (B-W numbering), Fig 4B-F (B-W numbering), Fig S7 (B-W numbering).

We have now added B-W and CGN numbering to all figures.

- Some claims are difficult to interpret or should be rephrased to avoid misinterpretations:

a) Page 3: 'D2R and D3R differ in brain distribution and signaling properties, and are both targeted by current antipsychotics and drugs for the treatment of neurological diseases (such as Parkinson's disease 15,16). Although agonists with some selectivity exist, treatments are still suboptimal due to a lack of selectivity and different effects originating from the activation of both receptors 17,18.' Especially in the case of antipsychotics, multi-target binding affinities have been postulated as a requirement for drug efficacy (see Roth, Nat Rev Drug Discov 2004) and in drugs like cariprazine, D2/D3 effects are believed to be required for antipsychotic action. Therefore this statement could be misleading and should be reformulated.

We have now clarified and the text currently states: “*Although agonists with some selectivity exist, new subtype selective molecules are likely to help understand their physiological role as well as providing leads for improved therapeutics. Indeed, selective activation of D₃R may yield neuroprotective effects in the treatment of Parkinson's disease, hence harboring potential in the treatment of neurodegeneration.*” (lines 101-103).

b) Page 4: please recheck 'allowing to separate two FOB02-04A and binding site conformations'.

We have now clarified the sentence to state: “*Positioning the ligand binding pocket at the center of the cryo-EM box improved the resolution at the D₃R extracellular region (Methods and Supplementary Fig. 3), and allowed us to classify two cryo-EM models containing two FOB02-04A conformations*”

c) Page 5: please recheck 'No major conformational changes are found in the D3R when

comparing it bound to pramipexole'.

We have now clarified the sentence to state: “*No major conformational changes were found at the D₃R when comparing its structure when bound to pramipexole (PDB 7CMU), PD128907 (PDB 7CMV), rotigotine (PDB 8IRT) or FOB02-04A (0.535 Å RMSD over 253 Ca in the pramipexole bound as example) aside from the ordering of the extracellular region of TM1 (see below).*”

d) Page 7: 'The cryo-EM density allowed modelling of the three bitopic components.' Does this mean 'The cryo-EM density allowed modelling of the three bitopic moieties.' or maybe 'The cryo-EM density allowed modelling of the three components of the bitopic ligand.'

We have now clarified the sentence to state *'The cryo-EM density allowed modelling of the three components of the bitopic ligand.'*

e) Page 8: 'This suggests a direct role in ligand-induced activation of the bitopic molecule.' Do the authors mean that this suggests a direct role of the bitopic molecule in ligand-induced receptor activation? In this sense, some of the mutations suggested above could provide further evidence to substantiate this claim.

Thank you for pointing this out, we have now clarified the sentence. The residue has been mutated to alanine and there is data comparing the effect on pramipexol, PD128907 and the bitopic agonist FOB02-04A to both wild-type and Y373^{7,43}A (within this manuscript and in other recent available papers), showing a differential effect between ligands (stronger detrimental impact for the bitopic ligand). It currently states: “*This suggests a role for this residue in the binding and/or function of the bitopic molecule to the receptor, in addition to its known role with D^{3.32}.*”

REVIEWER COMMENTS

Reviewer #1 (Remarks to the Author):

The revision of the manuscript addressed most of the comments of this referee. The reported Go/Gi selectivity was one of the most critical points that was adequately addressed by measuring the Go preference with the bitopic FOB02-04A. Furthermore, site directed mutagenesis was applied for the residues differed between GO and Gi located at the D3R:Galpha interface. The functional assays in HEK293T cells using BRET2 revealed the role of specific residues in the activation mechanism and Go coupling of FOB02-04A.

On the other hand, however, the revised version is still suggests that the SBP of FOB02-04A occupies a new site. As suggested by the original title of the manuscript (Structure of the dopamine D3 receptor bound to a bitopic agonist reveals a new specificity site in an expanded allosteric pocket) in the original version the authors claimed this site as an extended new pocket. Now they changed the title to „A bitopic agonist bound to the dopamine 3 receptor reveals a new selectivity site”, however, the novelty of the site has not been confirmed. In the rebuttal letter they wrote that „We have termed established SBP to such site to differentiate it from the new site which the bitopic ligand in the current manuscript describes, which extends the usable pocket to generate selective molecules.” However, they did not compare the binding mode of FOB02-04A to other bitopic compounds bound to dopaminergic and/or serotonergic receptors. The authors replied that „there are no experimental structures of bitopic ligands for other dopamine or serotonin receptors (at least that have proven to be bitopic, i.e. individual components can bind on their own), hence we cannot perform a structural comparison to such structures.” There are multiple issues with this statement. The first problem is that the authors defined bitopic ligands differently from that accepted in the literature (<https://pubs.acs.org/doi/10.1021/acs.jmedchem.6b01601> and <https://pubs.rsc.org/en/content/articlelanding/2023/cs/d3cs00650f>) : „Bivalent ligands are defined as consisting of two pharmacophores that simultaneously engage a combination of two allosteric and/or orthosteric binding sites. Dimeric ligands (ligands that bind to two separate receptors) and bitopic ligands (also termed dualsteric) are subclasses of bivalent ligands. The term bitopic ligand describes a molecule able to simultaneously bind to both the orthosteric and an allosteric binding site of a single receptor monomer. All bivalent ligands can be further categorized as homo- or heterobivalent, depending on whether the same pharmacophore is engaging both binding sites or not.” Consequently, the binding of individual components on their own is not the condition for bitopic compounds. In fact, the authors acknowledged that „several bitopic compounds with enhanced receptor selectivity have been developed for GPCRs”, however, most of the compounds reported in the cited papers are not in line with their suggested definition of „proven” bitopic nature. Furthermore, the selection of potential SBP binders in the paper describing FOB02-04A also did not consider the binding of individual components on their own (<https://pubs.acs.org/doi/10.1021/acs.jmedchem.9b00702>). Second, using the adequate definition of bitopic compounds, there are multiple compounds having structure in complex with

dopaminergic or serotonergic receptors. The authors may consider D2 inactive structures such as risperidone (6CM4), haloperidol (6LUQ) or spiperone (7DFP) and active 5-HT1A structures such as aripiprazole (7E2Z), IHCH-7179 (8JT6), Vilazodone (8FYL) or buspirone (8FYX). Interestingly, many of these compounds bind also to dopamine D3 receptors that suggest them as suitable comparator for analysing the location of the SBP found for FOB02-04A. Recently, Wacker et al. published <https://www.nature.com/articles/s41586-024-07403-2> the structural comparison of aripiprazole, buspirone and vilazodone in 5HT1A concluding that the SBP binding moiety of aripiprazole and vilazodone occupy the same SBP site (EBP1) while the SBP moiety of buspirone explored a new SBP site (EBP2). In the light of these findings the authors should investigate whether the SBP moiety of FOB02-04A is located at the equivalent of EBP1, EBP2 or it occupies a new site as stated in the manuscript. Consequently, the authors should compare their structure to that available in the literature and should demonstrate whether the indole moiety of FOB02-04A occupies a new site.

The authors made reasonable attempts to clarify the role deltaG94 in the selectivity of FOB02-04A that were finally inconclusive. Although these efforts are acknowledged, the selective binding of the compound to D3 receptors should be determined by specific interactions with this receptor that might not be available in D2. These interactions can be best characterized by binding experiments and therefore this reviewer disagrees that functionality represent better whether the compound is selective or not.

The revised version now contains different hypotheses on the role of the conformation B, however, neither of these were investigated in detail. The lack of a reasonable explanation makes this part of the manuscript very much speculative. Although this reviewer agrees that both the antagonist/weak agonist nature of conformation B or the existence of it is very low populated conformation with low affinity is challenging to detect on its own experimentally, however, MD simulations might add further information on its potential role. The authors should fix the internal degrees of freedom of FOB02-04A in conformation B and should perform MD simulations in both the active and inactive conformation of the receptor. These simulations might help understanding the relevance of conformation B.

Reviewer #2 (Remarks to the Author):

The authors addressed my concerns, but I have some minor issues.

1. R323, K345, I28, N194, V334, G350 and Y354 and related interactions mentioned in the paper should be shown in the Fig. 2c.
2. Line 190-197, interaction details between ICL2 of D3R and Gi/o proteins should be shown in the Fig. 2.

Reviewer #3 (Remarks to the Author):

The authors have now addressed most of my comments.

A few minor points:

- Fig. S1B does not mention that a GoA-TRUPATH biosensor was used.
- the manuscript uses 'dose-response curves' rather than 'concentration-response curves'. For cell-based assays, the term 'concentration-response curve' would be more precise because the actual concentrations are known. Doses would be more appropriate in animal experiments.
- lines 534-536: the current version is unclear to me. I assume the authors meant that they made mutations in the GaoA-RLuc8 construct?
- Figure 3E looks somewhat distorted
- Efficacy values in 3J are not readable

Reviewer #4 (Remarks to the Author):

Although I still believe that further mutagenesis at position Y365 could have helped to tease apart the contributions of FOB02-04A conformation B vs A towards receptor signalling, most of my comments have been satisfactorily addressed and I would agree with the publication of the current version of the manuscript.

Point-by-point response to reviewers (responses in blue)

We thank the reviewers for their comments and their positive feedback. We provide below a point-by-point answer to each of their comments and revisions:

Reviewer #1

The revision of the manuscript addressed most of the comments of this referee. The reported Go/Gi selectivity was one of the most critical points that was adequately addressed by measuring the Go preference with the bitopic FOB02-04A. Furthermore, site directed mutagenesis was applied for the residues differed between GO and Gi located at the D3R:Galpha interface. The functional assays in HEK293T cells using BRET2 revealed the role of specific residues in the activation mechanism and Go coupling of FOB02-04A.

We thank the reviewer for acknowledging our efforts.

On the other hand, however, the revised version still suggests that the SBP of FOB02-04A occupies a new site. As suggested by the original title of the manuscript (Structure of the dopamine D3 receptor bound to a bitopic agonist reveals a new specificity site in an expanded allosteric pocket) in the original version the authors claimed this site as an extended new pocket. Now they changed the title to „A bitopic agonist bound to the dopamine 3 receptor reveals a new selectivity site”, however, the novelty of the site has not been confirmed. In the rebuttal letter they wrote that „We have termed established SBP to such site to differentiate it from the new site which the bitopic ligand in the current manuscript describes, which extends the usable pocket to generate selective molecules.” However, they did not compare the binding mode of FOB02-04A to other bitopic compounds bound to dopaminergic and/or serotonergic receptors. The authors replied that „there are no experimental structures of bitopic ligands for other dopamine or serotonin receptors (at least that have proven to be bitopic, i.e. individual components can bind on their own), hence we cannot perform a structural comparison to such structures.” There are multiple issues with this statement. The first problem is that the authors defined bitopic ligands differently from that accepted in the literature (<https://pubs.acs.org/doi/10.1021/acs.jmedchem.6b01601> and <https://pubs.rsc.org/en/content/articlelanding/2023/cs/d3cs00650f>): „Bivalent ligands are defined as consisting of two pharmacophores that simultaneously engage a combination of two allosteric and/or orthosteric binding sites. Dimeric ligands (ligands that bind to two separate receptors) and bitopic ligands (also termed dualsteric) are subclasses of bivalent ligands. The term bitopic ligand describes a molecule able to simultaneously bind to both the orthosteric and an allosteric binding site of a single receptor monomer. All bivalent ligands can be further categorized as homo- or heterobivalent, depending on whether the same pharmacophore is engaging both binding sites or not.” Consequently, the binding of individual components on their own is not the condition for bitopic compounds. In fact, the authors acknowledged that „several bitopic compounds with enhanced receptor selectivity have been developed for GPCRs”, however, most of the compounds reported in the cited papers are not in line with their suggested definition of „proven” bitopic nature. Furthermore, the selection of potential SBP binders in the paper describing FOB02-04A also did not consider the binding of individual components on their own (<https://pubs.acs.org/doi/10.1021/acs.jmedchem.9b00702>). Second, using the adequate definition of bitopic compounds, there are multiple compounds having structure in complex with dopaminergic or serotonergic receptors. The authors may consider D2 inactive structures such as risperidone (6CM4), haloperidol (6LUQ) or spiperone (7DFP) and active 5-HT1A structures such as aripiprazole (7E2Z), IHCH-7179 (8JT6), Vilazodone (8FYL) or buspirone (8FYX). Interestingly, many of these compounds bind also to dopamine D3 receptors that suggest them as suitable comparator for analysing the location of the SBP found for FOB02-04A. Recently, Wacker et al. published <https://www.nature.com/articles/s41586-024-07403-2> the structural comparison of aripiprazole, buspirone and vilazodone in 5HT1A concluding that the SBP binding moiety of aripiprazole and vilazodone occupy the same SBP site (EBP1) while

the SBP moiety of buspirone explored a new SBP site (EBP2). In the light of these findings the authors should investigate whether the SBP moiety of FOB02-04A is located at the equivalent of EBP1, EBP2 or it occupies a new site as stated in the manuscript. Consequently, the authors should compare their structure to that available in the literature and should demonstrate whether the indole moiety of FOB02-04A occupies a new site.

We thank the reviewer for the clarification and we apologize for the misunderstanding, we do agree that the novelty of the ligand binding site should be more clearly demonstrated and shown in the manuscript.

The D₃R/D₂R selectivity site is currently composed of G94 and H29. The novelty of the site at the level of the D₃R is clear since TM1 has never been ordered in any published structure and there are no ligands binding at G94 position either. In order to confirm the novelty at the level of other aminergic receptors with bitopic molecules we have aligned all structures suggested by the reviewer and shown the relative position of the bitopic ligands on the D₃R structure, this shows that no other ligand occupy this site. The closest binding mode would be that of vilazodone on the serotonin 5HT_{1A} receptor, where the terminal amide group of vilazodone is close to N100 in ECL1 (the G94 equivalent), but out of H-bonding distance. This discussion has been included in the manuscript, inside section “Diversity of the SBP2-ECL1-1 in other aminergic receptors”, as well as adding a new panel to main Figure 4 showing the relative position of all described ligands at the D₃R.

The authors made reasonable attempts to clarify the role deltaG94 in the selectivity of FOB02-04A that were finally inconclusive. Although these efforts are acknowledged, the selective binding of the compound to D₃ receptors should be determined by specific interactions with this receptor that might not be available in D₂. These interactions can be best characterized by binding experiments and therefore this reviewer disagrees that functionality represent better whether the compound is selective or not.

We apologize for the confusion, we do agree that radioligand binding experiments are a useful tool for measuring the selectivity of ligand binding, however, we would like to highlight that these assays would be of lesser use in the current system. The selective residues for FOB02-04A between D₃R and D₂R are G94 and H29. These residues are specific for the D₃R, but also for conformation A. Mutating those residues are likely to promote the binding in conformation B, and hence, observing ligand binding in radioligand binding assays on the mutants would not inform about the role of H29 or G94 in ligand selectivity, since although the ligand could bind, it does not result in receptor activation, as seen in functional assays done with ΔG94 variant. In this case, the ligand is functionally selective but might not be binding selective. Hence, the radioligand binding assays would not provide key information about the role of G94 and H29 in binding selectivity.

The revised version now contains different hypotheses on the role of the conformation B, however, neither of these were investigated in detail. The lack of a reasonable explanation makes this part of the manuscript very much speculative. Although this reviewer agrees that both the antagonist/weak agonist nature of conformation B or the existence of it is very low populated conformation with low affinity is challenging to detect on its own experimentally, however, MD simulations might add further information on its potential role. The authors should fix the internal degrees of freedom of FOB02-04A in conformation B and should perform MD simulations in both the active and inactive conformation of the receptor. These simulations might help understanding the relevance of conformation B.

This is indeed a great suggestion, and another reviewer suggested it in the previous round. Hence, these MD simulations have already been done. We performed MD simulations to understand whether conformation B was stabilizing the inactive conformation more than conformation A. For this purpose, both conformations were docked into the crystal structure of the inactive-state D₃R and subject to MD simulations (see Figure 1 below). Although worth trying, this experiment did not show clear trends; however, we would like to point out that this strategy would only show results if conformation B were to be an inverse agonist and will not show a trend if the bitopic ligand is a very weak agonist or an antagonist.

Figure 1. Distribution of binding poses of bitopic FOB02-04A in D₃R-inactive-state complex. (A) Predicted binding poses of bitopic FOB02-04A with inactive-state dopamine D₃ receptor (D₃R). D₃R is shown in cyan, the docked conformation A (red) and conformation B (green) of FOB02-04A are shown as capped sticks, and intramolecular interactions are shown as black dashes lines. Within the binding pocket, residues interacting with conformations A and B are depicted as sticks, with E90^{2.65} and Y365^{7.35} highlighted in bold. Arrows indicate distances for assessing bitopic FOB02-04A's binding pose distribution between conformations A and B, specifically from E90^{2.65}'s carboxyl group in D₃R to FOB02-04A's indole atom type N5 (red) and from Y365^{7.35}'s 4-hydroxyphenyl moiety in D₃R to the phenyl ring of FOB02-04A's 1H-indole-2-carboxamide SP (green), the closest distances are depicted as well; (B) Hydrogen bond interaction dynamics between D₃R's E90^{2.65} carboxyl group and FOB02-04A's indole atom N5 (depicted in brown pallet) compared with proximity distance between 4-hydroxyphenyl moiety of D₃R's Y365^{7.35} and phenyl ring of 1H-indole-2-carboxamide SP of FOB02-04A's (shown in green palette) for **ConformationA-bound D₃R**; (C) Hydrogen bond interaction dynamics between D₃R's E90^{2.65} carboxyl group and FOB02-04A's indole atom N5 (depicted in brown pallet) compared with proximity distance between 4-hydroxyphenyl moiety of D₃R's Y365^{7.35} and phenyl ring of 1H-indole-2-carboxamide SP of FOB02-04A's (shown in green palette) for **ConformationB-bound D₃R**.

Additionally, I would like to highlight that we made significant efforts during the last round of revisions to understand whether conformation B of the FOB02-04A is really an antagonist. For this purpose we:

- Performed competition assays using BRET cellular assays where we used the ΔG94 variant, aiming to force bitopic binding into conformation B, and then aimed to compete quinpirole or titrated quinpirole in the presence of the bitopic molecule FOB02-04A.
- Used fluorescently labeled dopamine to perform similar competition assays on the ΔG94 D₃R variant expressed in HEK293T cells.

Unfortunately, working with the ΔG94 variant is technically challenging and we did not obtain conclusive results in either direction when performing either type of experiments.

Although we do agree this last section is more speculative we thought it was worth showing the computational and the experimental structure showing the two conformations, which regardless of the nature of the conformation (either an antagonist or a low populated non-selective conformation), it highlights that the flexibility of large ligands can result in binding flexibility that can hinder the development of selective and full agonist molecules, which is a major challenge in developing subtype selective agonists for the D₃R.

Reviewer #2:

The authors addressed my concerns, but I have some minor issues.

1. R323, K345, I28, N194, V334, G350 and Y354 and related interactions mentioned in the paper should be shown in the Fig. 2c.

We have tried to add these residues in Fig 2C, unfortunately adding all those residues to what was Fig 2C before, would require a zoom out that would lose the details shown initially. For this reason, we have now modified Fig 2 as a whole so as to highlight all these residues in as many panels as possible so it is clear where they are relative to the structure.

2. Line 190-197, interaction details between ICL2 of D3R and Gi/o proteins should be shown in the Fig. 2.

We have now modified Fig 2 so as to show the interaction details between ICL2 of D₃R and G_{i/o} proteins as a new panel Fig 2C as suggested by the reviewer.

Reviewer #3:

The authors have now addressed most of my comments. A few minor points:

- Fig. S1B does not mention that a GoA-TRUPATH biosensor was used.

We apologize for missing this, we have now added that G_{OA}-TRUPATH biosensor was used in the figure legend of Fig S1B.

- the manuscript uses 'dose-response curves' rather than 'concentration-response curves'. For cell-based assays, the term 'concentration-response curve' would be more precise because the actual concentrations are known. Doses would be more appropriate in animal experiments.

We agree with the reviewer that 'concentration-response curves' is a more accurate term, we have now replaced 'dose-response curves' with 'concentration-response curves'.

- lines 534-536: the current version is unclear to me. I assume the authors meant that they made mutations in the G_{αO}A-RLuc8 construct?

The reviewer interpretation is correct, we have now modified the text to enhance clarity.

See line 534-535. "The G_{αO}A mutants to study the G_{i/o} selectivity of the D₃R were done in the G_{αO}A-RLuc8 construct."

- Figure 3E looks somewhat distorted

We have modified Figure 3 to ensure high quality.

- Efficacy values in 3J are not readable

We have removed efficacy values from figure 3J, they are challenging to show clearly in the figure and we have left these values in Supplementary material.

Reviewer #4:

Although I still believe that further mutagenesis at position Y365 could have helped to tease apart the contributions of FOB02-04A conformation B vs A towards receptor signalling, most of my comments have been satisfactorily addressed and I would agree with the publication of the current version of the manuscript.

We thank the reviewer for the positive comment.

REVIEWERS' COMMENTS

Reviewer #1 (Remarks to the Author):

The only critical point of the revised manuscript was the novelty of the observed binding site that is now addressed adequately. The selectivity site described in this work is in fact new for D3 since the binding mode of other bitopic ligand was not investigated previously. It is important, however, that the site itself is also present in 5HT1A receptor as shown in panel B of the revised Figure 4.

Overlapping the relevant binding modes revealed that this compound binds to the established SBP of Class A GPCRs. Although there are no data on 5HT1A binding of FOB02-04A, it is more than likely that the compound binds to this serotonergic receptor.

All other comments were addressed, and the revised version is suitable for publication.

Point-by-point response to reviewers (responses in blue)

Reviewer #1:

The only critical point of the revised manuscript was the novelty of the observed binding site that is now addressed adequately. The selectivity site described in this work is in fact new for D₃ since the binding mode of other bitopic ligand was not investigated previously. It is important, however, that the site itself is also present in 5HT_{1A} receptor as shown in panel B of the revised Figure 4. Overlapping the relevant binding modes revealed that this compound binds to the established SBP of Class A GPCRs. Although there are no data on 5HT_{1A} binding of FOB02-04A, it is more than likely that the compound binds to this serotonergic receptor.

All other comments were addressed, and the revised version is suitable for publication.

We thank reviewer 1 for the comments and positive feedback. We are glad the new figure made the novelty of the new binding site clearer. Indeed, some overlap of the secondary binding site is present between the 5HT_{1A} receptor and the D₃R, however, since the pocket shape and residues therein are not conserved between these receptors it is not obvious to say that the secondary pharmacophore will bind to both receptors, let alone the entire ligand. Overall we appreciate the comments from the reviewer who have contributed significantly to improve the manuscript.

ADDITIONAL COMMENT FROM THE AUTHORS

The authors would like to report an error and its correction during the execution of the experiments performed in the previous round of corrections. In this experiments we performed a characterization of the receptor:G protein interface key interactions, which involved the potency characterization of 5 mutants as shown in figure 2f. We identified a consistent 10x dilution error that affects all data points for potency equally, yielding an underestimation of the potency values. This does not change the overall message or the description in the manuscript as explained below but we have corrected such error.

We initially described that all mutants had significant effect in the receptor:G protein interface, but one of the residues had the most impact, which is the residue we have dedicated more discussion, MD simulations and figure panels to (i.e. Fig 2b). With the change in values the less prominent residues are now not statistically significant, while the key interaction which we devote all of our effort and discussion remains significant. Hence, the correction does not affect the figure panel message. However, we would like to state that we modified panel Fig 2F and associated Source Data values to correct for this. The text also has a minor correction in one sentence to account for this:

Before (lines 203-205):

Overall, all mutations had an impact on the G_O protein coupling with residue G350^{G.H5.22} located at the C-terminal $\alpha 5$ having he most impact and hence the most determinant of the D₃R:G_O selectivity (Fig. 2F).

After (lines 203-205):

Overall, only mutation of residue G350^{G.H5.22}, located at the C-terminal $\alpha 5$, had a significant impact on its own and hence this residue is the most determinant of the D₃R:G_O selectivity (Fig. 2F).